# Lung250M-4B: A Combined 3D Dataset for CT- and Point Cloud-Based Intra-Patient Lung Registration

**Fenja Falta**
Institute of Medical Informatics
University of Lübeck
`fenja.falta@student.uni-luebeck.de`

**Christoph Großbröhmer**
Institute of Medical Informatics
University of Lübeck
`c.grossbroehmer@uni-luebeck.de`

**Alessa Hering**
Departments of Imaging
Radboud University Medical Center, Nijmegen

**Alexander Bigalke**
Institute of Medical Informatics
University of Lübeck

**Mattias P. Heinrich**
Institute of Medical Informatics
University of Lübeck

## Abstract

A popular benchmark for intra-patient lung registration is provided by the DIR-LAB COPDgene dataset consisting of large-motion in- and expiratory breath-hold CT pairs. This dataset alone, however, does not provide enough samples to properly train state-of-the-art deep learning methods. Other public datasets often also provide only small sample sizes or include primarily small motions between scans that do not translate well to larger deformations. For point-based geometric registration, the PVT1010 dataset provides a large number of vessel point clouds without any correspondences and a labeled test set corresponding to the COPDgene cases. However, the absence of correspondences for supervision complicates training, and a fair comparison with image-based algorithms is infeasible, since CT scans for the training data are not publicly available. We here provide a combined benchmark for image- and point-based registration approaches. We curated a total of 248 public multi-centric in- and expiratory lung CT scans from 124 patients, which show large motion between scans, processed them to ensure sufficient homogeneity between the data and generated vessel point clouds that are well distributed even deeper inside the lungs. For supervised training, we provide vein and artery segmentations of the vessels and thousands of image-derived keypoint correspondences for each pair. For validation, we provide multiple scan pairs with manual landmark annotations. Finally, as baselines on our new benchmark, we evaluate several image and point cloud registration methods on the dataset.

## 1 Introduction

Image registration continues to be an important and challenging task in medical computer vision. A large part of recently developed methods for registration incorporates deep learning [13]. With advances in geometric deep learning, point cloud-based registration approaches have become increasingly more prevalent and compete well with image-based approaches, especially in terms of efficiency and robustness [39]. However, as deep learning is data-driven, its development is highly dependent on the existence of appropriate datasets and benchmarks.

37th Conference on Neural Information Processing Systems (NeurIPS 2023) Track on Datasets and Benchmarks.

As the results of the Learn2Reg benchmark [21] indicate, 3D lung registration remains challenging for learning-based approaches, and contrary to other image analysis tasks, conventional optimisation-based approaches remain state-of-the-art (SOTA). We believe this is in part due to the limitation of available lung datasets, which typically lack both sample size and range of motion, thereby restricting further advances through deep learning methods.

With Lung250M-4B we propose a combined dataset consisting of Lung4B, 248 paired CT scans from 124 patients (with approximately 4 billion voxels in total), and Lung250M, with corresponding lung vessel point clouds with approximately 250 million 3D points and features. This dataset comprises multi-centric data from different examination types with more cases than comparable datasets and furthermore enables a fair comparison between image- and point cloud-based registration approaches. The dataset is available at `https://cloud.imi.uni-luebeck.de/s/s64fqbPpXNexBPP`.

**Clinical Motivation for Lung Registration**  3D registration in general often deals with the alignment of scenes and (deformable) objects therein, e.g. for shape completion and scene flow estimation. 3D medical registration, in particular, focuses on finding detailed anatomical or functional correspondences. This could be either for different time points of the same patient or across a population for shape analysis. While subtle small deformations can be reliably estimated with a range of voxel- and intensity-based approaches [2, 29], larger lung motion is more difficult to capture and hence graph- or point-cloud based alignment has become more prominent [48, 47].

Finding correspondences between different respiratory levels of lung CT within the same patient is vital for the diagnosis and characterisation of chronic obstructive pulmonary disease (COPD) [14]. In COPD local airflow is limited due to small airway disease or emphysema, and accurate nonlinear displacement fields provide an accurate means for calculating local volume change (and hence areas of limited airflow) based on the Jacobian determinant [45]. Two other important areas of lung registration are: estimation of tumour motion from 4D-CT in radiotherapy planning [54] and identification and tracking of lung nodules for screening and diagnostics of lung cancer [22, 40].

**Benefits of Learning on Point Clouds**  In recent years there has been an increase in research regarding point cloud learning, which can also be seen in medical research (e.g. [52, 61]). The main difference between point cloud and image data lies in discarding image (intensity) information that is either redundant or not necessary for the task, which can be advantageous. First, focusing on only relevant sparse information makes algorithms more efficient and reduces the risk of overfitting to subtle irrelevant appearance details. Second, graph-based networks excel at learning representations that are more closely aligned to underlying anatomical concepts (e.g. vessel tree topology) and invariant to certain operations (e.g. local rotations). Third, for sensitive data, in particular in the medical domain, the public sharing of images is often not possible due to patient privacy. Datasets thus often remain private. Point clouds generated from CT scans, however, hold significantly less information that makes the patient identifiable than CT scans themselves. Hence, point cloud-based computer vision methods preserve anonymity and enable access to more data.

## 2   Related Work

**Medical registration models.**  The majority of methods address 3D image registration on a dense grid of control points (e.g. B-splines [37]) and minimise a joint cost function of image dissimilarity (e.g. normalised cross-correlation [1]) using a discrete [47], continuous or stochastic optimiser [29]. Deep learning-based approaches follow either a multi-scale architecture [38], use correlation windows for a set of discretised displacements [50, 17] or focus on more subtle diffeomorphic deformations [2]. While there are certain approaches, including spherical demons [60] or graph- and learning-based methods [49] that address cortical matching on the surface of brains [28], only few works addressed medical registration on the basis of purely geometric point clouds. Standard classical approaches like Iterative Closest Point [5] or Coherent Point Drift [43] do naturally apply to the problem but were shown insufficient for registering highly complex structures in [48]. [27] extended the Bayesian Coherent Point Drift to the registration of multiple organ surface point clouds. As a learning-based method, [4] proposed a PointNet-based model for prostate surface registration, trained in an unsupervised manner with the Chamfer distance. [15] performed keypoint-based lung registration by combining a graph network for geometric feature learning with loopy belief propagation for differentiable optimisation, which, however, required keypoint correspondences for

supervision. Finally, [48] combined the deep learning-based PointPWC-Net [56] with modules from optimal transport theory for pre-alignment and fine-tuning to register high-resolution lung vessel trees. Training was performed on synthetic deformations.

**Medical registration datasets.** Starting from the retrospective intermodality brain registration evaluation [55] that considered only rigid alignment, numerous challenges and public datasets have been proposed for medical image registration, including lung registration [42] and multi-modality MRI to ultrasound registration [58]. A comprehensive challenge that also considered certain aspects of learning-based method is Learn2Reg [21], which comprises several complementary tasks and has compared dozens of SOTA approaches. Nevertheless, apart from OASIS - an inter-subject brain mapping tasks - all mentioned datasets comprise 50 or fewer scans and are thus not ideally suited for deep learning approaches. Public benchmarks for point cloud-based medical image registration are rare since most above point cloud registration methods were evaluated on in-house datasets [4] or the keypoints were derived with private tools and not made publicly available [27]. The only public medical point cloud dataset is the PVT dataset [48], providing 1000 unlabeled inhale-exhale pairs of lung vessel trees along with 10 labeled test cases, each annotated with 300 landmark correspondences.

**3D point cloud registration benchmarks.** Beyond medical applications, 3D scene flow and point cloud registration are active research fields in computer vision and graphics, where diverse public benchmarks are available. The Flying Things 3D dataset [34] features rigid motions of synthetic 3D object models. Similarly, synthetic 3D models from the ModelNet40 [57] dataset are rigidly transformed and subsampled for partial-to-partial registration. Piecewise rigid motions on real point cloud data are particularly present in datasets from autonomous driving, including the KITTI Scene Flow [36, 35] and Waymo [51] datasets. Unlike (piecewise) rigid motions, the Faust [7] and Dynamic Faust [8] datasets feature deformable human motions. Moreover, the DeformingThings4D [32] and derived 4DMatch datasets [31] provide frames from animated motion sequences of humans and diverse animals. Beyond the different (medical) scope, our dataset differs from all above benchmarks in two decisive technical aspects. 1) Our point clouds do not only represent surfaces but complex interior structures, making the registration particularly challenging and impeding unsupervised learning with similarity metrics. 2) We jointly provide image and point cloud data, enabling a fair comparison of image and point-based methods and facilitating the development of methods that incorporate both types of data.

**Contribution** Lung250M-4B substantially extends prior lung registration datasets by a factor of 3-4× and provides paired publicly available CT scans and high-resolution point clouds with the aim to bring together research from 3D vision and medical imaging. We enhance the raw data substantially by adding a large amount of automatic high-quality correspondences on both images and point clouds to provide stronger supervision with the potential to ease and/or improve training. We also provide initial benchmark evaluation for several baselines: image- and point cloud-based, learning- and optimisation-based, unsupervised/self-supervised and supervised.

## 3   Dataset Generation

We curated 248 publicly available 3D CT scans of 124 patients from various medical centres, with image acquisition ranging from in- and expiratory breath-hold CT over maximum and minimum volume scans of 4DCT scans to base and follow-up scan pairs at different time points. Data are acquired from various scanners with various resolutions and slice thicknesses, and show a wide variety regarding occurring pathologies. An overview of source datasets used for curation can be found in Tab. 1, while detailed information regarding acquisition centres, subject pathologies and image metadata is provided in the supplementary material.

The **DIR-LAB COPDgene** dataset[9], published in 2013, includes ten randomly picked pairs of CT-breath hold examinations obtained as part of the COPDGene study (NCT00608764) to evaluate genetic factors for the development of COPD in smokers. Each scan pair is associated with manually annotated landmarks (n = 300) and an inter-observer landmark localisation error, rendering the dataset well-suited for evaluation of deformable lung registration algorithms. The data are not released under

Table 1: Overview of the data set curation. We include a total of 248 lung scans from different examination types (BH: Breath-Hold, LS: Longitudinal Scans, 4DCT) from a total of 6 public and freely available data collections in our dataset. The last column shows the mean change (and its standard deviation) of the lung volume between the two images of a scan pair.

| Source Dataset | # Included Scan Pairs | # Annotated Cases (# Landmarks/case) | Examination Type | Mean Lung Volume Change in ml ($\pm$std) |
| --- | --- | --- | --- | --- |
| DIR-LAB COPDgene | 10 | 10 (300) | BH | $1770 \pm 708$ |
| Empire10 | 12 | 2 (100) | BH,4DCT | $1950 \pm 1274$ |
| L2R-LungCT | 30 | 3 (100) | BH | $2155 \pm 428$ |
| TCIA-NSCLC | 20 | - | 4DCT | $426 \pm 201$ |
| TCIA-Ventilation | 20 | 2 (100) | BH | $1137 \pm 559$ |
| TCIA-NLST | 32 | 10 (100) | LS | $766 \pm 505$ |

a standard licence, but can be obtained after filing an online request form, with the restriction that the source be acknowledged in case of publication[1].

The **EMPIRE10** dataset was released as part of a MICCAI 2010 workshop for a representative benchmarking of lung registration algorithms [42]. It consists of a total of 30 scan pairs intended to represent a wide range of clinical applications. From this set, we choose the subsets "Insp-Exp" (8 cases) and "4D" (4 cases) to ensure sufficient lung volume change between scans. The dataset is not released under a standard licence, but can be obtained from the EMPIRE10 challenge website[2].

**L2R-LungCT** is one of various medical registration tasks included in the Learn2Reg challenge [21] with the goal of registration of in- and expiratory lung CT scans. In total, 30 scan pairs from this task are publicly available[3], published under *CC-BY-4.0*. Only scan pairs that a) were breath-hold examinations, b) had sufficient lung coverage in both images, and c) had more than 300 slices were considered. All identifying patient data were anonymized. To determine registration accuracy, manual annotations for 3 validation cases have been released, containing 100 landmarks generally located at vessel, airways and parenchyma bifurcations and surfaces.

The National Lung Screening Trial (**NLST**, NCT00047385) [44, 53, 10] compared the application of low-dose CT to chest radiography in high-risk lung cancer patients in the USA. More than 26,000 of the 75,000 low-dose CTs are freely available through the Cancer Imaging Archive under a *CC-BY 4.0* licence, making it one of the largest medical imaging resources available[4]. The imaging protocol did not provide for a defined breathing state, so the baseline and follow-up scans may have been acquired at different times of the respiratory cycle. Our goal is to include only scan pairs with a sufficiently large lung motion in our dataset. Therefore, a larger set of scan pairs including 281 subjects was initially considered but narrowed down to 22 cases with a minimum lung volume change of 380 ml. In addition, we included additional 10 scan pairs used in the Learn2Reg challenge, which were released with manual landmark annotations. Because we have no influence on the selection of validation cases, the mean change in lung volume for validation scan pairs is 247 ml.

The **TCIA-Ventilation** dataset[12, 10] contains image data from a study comparing scans during breath-hold examinations, free-breathing examinations, and lung ventilation examinations using Galligas PET with a total of 20 participants [11]. The study was approved by the local ethics committee and registered with the Australian New Zealand Clinical Trials Registry (ACTRN12612000775819). All data have been licenced under *CC-BY-4.0* and can be accessed via TCIA[5]. We retrieve all (20) breath-hold inhale/exhale scan pairs from this dataset.

---

[1]`https://med.emory.edu/departments/radiation-oncology/research-laboratories/deformable-image-registration/downloads-and-reference-data/copdgene.html`

[2]`https://empire10.grand-challenge.org/Download/`

[3]`https://zenodo.org/record/4279348`

[4]`https://wiki.cancerimagingarchive.net/display/NLST/National+Lung+Screening+Trial`

[5]`https://wiki.cancerimagingarchive.net/pages/viewpage.action?pageId=125600096`

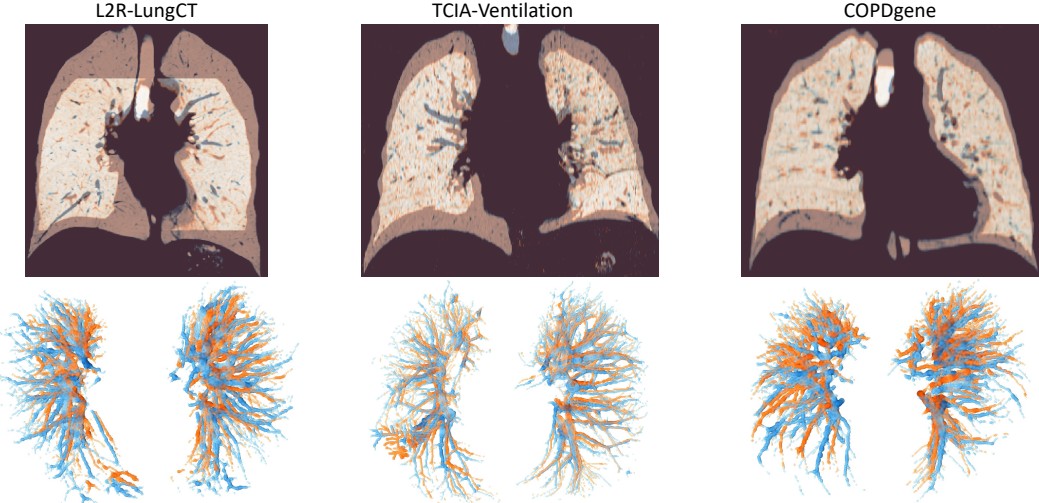

Figure 1: Visualization of three sample cases from our dataset. For each case, we show an overlay of fixed (blue) and moving (orange) CT slices (top) and skeletonised 3D lung vessel trees (bottom).

The **TCIA-NSCLC** data collection[23, 24, 3, 46, 10] includes CT scans of 20 patients who received radiation for the treatment of non-small cell lung cancer, released under *CC-BY-3.0* that can be obtained via TCIA[6]. Prior to image-guided radiation therapy, a 4DCT was acquired from each patient. We include all scans with maximal inspiration or expiration phases, resulting in 20 image pairs.

**Additional Validation Cases**   The proposed dataset consists of 6 subsets, of which only NLST, Learn2Reg, and COPDgene already have scan pairs with manual annotations for in-dataset registration validation. Students with a background in medicine further annotated two additional cases each for the TCIA-Ventilation and EMPIRE10 subsets, following the procedure described in [42] using the freely available software isimatch[7] [41] with 100 distinct landmarks on each scan.

**Ethical Discussion**   The presented Lung250M-4B dataset is a composite of previously published de-identified data collections. Hence, the ethical risk assessment of re-releasing the data is particularly linked to their original publication. We chose to use only data whose associated studies had been published in reputable peer-reviewed journals. In addition, the benchmark datasets EMPIRE10 and L2R-LungCT were reviewed and accepted by the MICCAI conference organisers and are widely used in the medical image registration community. At TCIA, providing medical data can only be done when abiding by the ethical code of conduct, which explicitly requires permission from a local institutional review board if needed. Therefore, we do not see any unassessed ethical risk in publishing the images and information extracted from the dataset. We remark, however, that the occurring pathologies do not reflect the general population (cf. Suppl. Tab. 1) and our dataset is only intended for research purposes and not for clinical usage. Because the dataset we present aims to establish a fair basis for evaluation between image- and point-cloud-based lung registration, we further consider the publication as an important long-term contribution to anonymity-preserving point cloud research in the medical domain.

## 4   Methods

Our methods for dataset generation comprise steps of image analysis, point cloud and graph processing as well as sparse deformable keypoint matching.

**Image Processing**   To ensure sufficient similarity between the curated data from different datasets, we employ joint preprocessing steps. For each image, a lung segmentation has been generated via a

---

[6]https://wiki.cancerimagingarchive.net/pages/viewpage.action?pageId=21267414
[7]https://www.isi.uu.nl/research/software/isimatch/

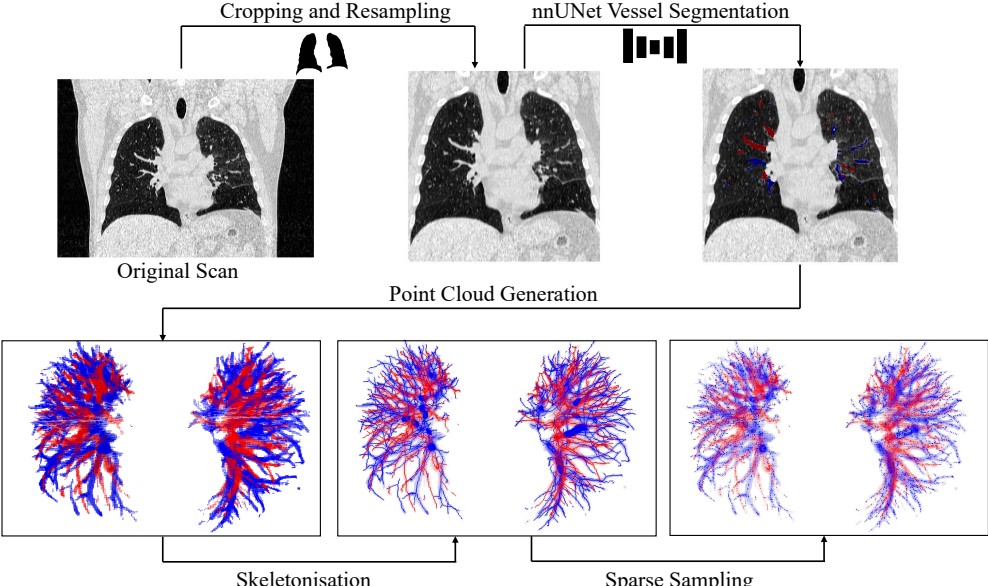

Figure 2: Image preprocessing and subsequent generation of point clouds. First, the original scan gets resampled and cropped around the lung mask. Then, arteries and veins are segmented via a trained nnUNet. Based on these, vessel point clouds are generated, which are subsequently skeletonised and reduced to obtain a sparse yet representative point cloud.

trained nnUNet [25] prediction. The network has been trained on 500 publicly available lung CT scans from a variety of different centres and scanners with 2 mm resolution. Since resolution between the original images differ greatly, images and predicted lung masks are resampled to isotropic 1 mm resolution using the public c3d toolbox [8] and, if necessary, reoriented to RAI orientation. The scans are then cropped around the lung masks with a distance of 10 mm to the lung segmentation.

To enable a reproducible pre-processing on further data, e.g. from a private dataset, all source code, tools, scripts and models to perform these steps are provided (cf. Supplementary Material).

**Vessel and Point Cloud Extraction**   We trained another nnUNet [25] to predict vessel segmentations of CT scans, divided into vein and artery labels. To train the network, we used the Parse22 challenge [33] dataset, consisting of 100 training CT scans with ground truth artery segmentations. This dataset is resampled and cropped as described above to ensure applicability of the segmentation network on the registration data. To obtain full vessel segmentations for the training data, we employed a self-supervised vessel enhancement method as described in [26]. We used this vessel segmentation as a base and then marked voxels of the segmentation as either vein or artery voxel. This is done by overlaying the vessel segmentation with the ground truth artery segmentations, however dilated by 1 mm in each direction to ensure consistent labeling of the whole vessel. Mirroring has been disabled as data augmentation to provide the network with more geometric information to distinguish arteries and veins. The nnUNet achieves a validation Dice on the Parse22 dataset (with ground truth semantic segmentations generated as described) of 81.4% for the vein and 84.9% for the artery. After inference on the registration data, segmentation predictions have been masked to the inside of the lung again to eliminate potential artefacts.

Based on the predicted vessel segmentations, we generated point clouds covering the full segmentation information by sampling each labelled voxel coordinate after interpolation by a scale factor of 2. This results in point clouds with .5 mm spacing and over 300k 3D points (1.1M on average) for each case. With these, different sampling strategies can be evaluated. To obtain points that are sampled with a similar density in both shallower and deeper parts inside of the lung, we employ a skeletonisation of the vessel segmentations following the method described in [30], resulting in 30'000 points on average per case. Since many graph networks require a fixed number of points in every cloud, we also

---

[8] http://www.itksnap.org/pmwiki/pmwiki.php?n=Convert3D.Convert3D

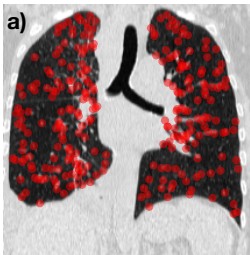 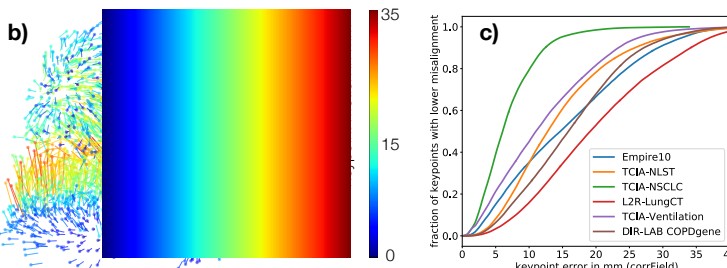

Figure 3: Generation of keypoint correspondences: a) geometric interest points on slice of fixed CT scan, b) 3D motion vectors of **corrField**, c) distribution of initial keypoint errors across datasets.

provide a sparse subset of these skeleton point clouds. For this purpose, we create a kNN-graph from the skeleton point clouds and employ a Gaussian density estimation followed by non-local maximum suppression to sample either exactly 8'192 or 16'384 voxel coordinates for each cloud. For all points, we provide the label of the voxel (artery resp. vein) and the vessel radius as features. A visualisation of the preprocessing and point cloud extraction pipeline can be seen in Fig. 2.

**Automatic Correspondences**   To aid learning-based methods, we computed a large number of highly accurate one-to-one correspondences across matching anatomical details of each scan pair using the public **corrField** method[9] [20]. Based on image-derived features and geometric interest points found within the lungs, it builds a graph for the inspiration scan and evaluates a combinatorial cost function for a large discretised set of potential displacements to match the most likely points in the expiration scan. At the same time, the correspondence field should be spatially smooth, which is achieved by a Markov random field optimisation. Our training set comprises 700'861 of these correspondence labels, roughly 6'750 per case. Visual details on the distribution of keypoints and correspondences for a particular patient as well as statistics on the initial alignment error for each subset of our data is shown in Fig. 3.

**Dataset Composition**   For each of the 248 scans we provide the following data: 1) Preprocessed CT scans (cropped and resampled), 2) corresponding lung masks and vein/artery-segmentations as .nii.gz-Files, 3) corrField keypoint correspondences as .csv-Files, 4) point clouds of skeletonised vessels (full and sparsely sampled) with corresponding features (label and vessel radius) as .pth-Files and 5) validation landmarks for the annotated scans as .pth-Files (only for a subset).

## 5   Experiments

We evaluate different registration methods on our dataset, including conventional registration without learning, image-based deep learning (DL) methods, and point cloud-based DL methods. As detailed above, the DIR-LAB COPDgene dataset with its 20 CT scans of 10 COPD patients at breathhold inspiration and expiration serves as test set for the proposed benchmark. For each scan pair, 300 manual landmarks with high inter-rater agreement are available for quantifying registration accuracy. Since the scan pairs have already been cropped based on the lung masks (bounding boxes) the initial registration error of the COPDgene dataset is lowered after this implicit translation alignment from 23.36 mm to 16.25 mm. First, we evaluate the quality of the provided pseudo-labels, by extrapolating the sparse **corrField** keypoint correspondences [20] of the test images to the manual landmark coordinates using a thin-plate-spline transform. This results in an accuracy of 1.45 mm. Next, we apply the conventional 3D image registration method **deeds** [19] that was designed for deformable lung alignment and performed best in a comparative study of six SOTA methods for abdominal registration [59]. The only modification was to supply 3D scans where everything outside the lungs is masked out. It runs for about 10 minutes on a 28-core CPU and yields a target registration error (TRE) of 1.53 mm.

Since the very popular baseline of VoxelMorph [2] does not achieve satisfactory performance on highly deformable 3D registration, we employ the **VoxelMorph+** adaptation of [18] that adapts the

---

[9]`https://grand-challenge.org/algorithms/corrfield/`

Table 2: Quantitative results on the COPD test cases of image-based (left) and point-based (right) methods, reported as mean TRE and 25/50/75% percentiles in mm. IO: Instance optimisation

| Method | TRE | 25% | 50% | 75% | Method | TRE | 25% | 50% | 75% |
|---|---|---|---|---|---|---|---|---|---|
| initial | 16.25 | 10.14 | 15.94 | 21.76 | initial | 16.25 | 10.14 | 15.94 | 21.76 |
| corrField | **1.45** | 0.83 | 1.23 | **1.77** | CPD | 3.13 | 1.51 | 2.28 | 3.58 |
| deeds | 1.53 | 0.83 | 1.22 | 1.79 | CPD w/ labels | **2.59** | **1.36** | **2.01** | **3.16** |
| VM+ w/o IO | 6.53 | 3.38 | 5.82 | 8.50 | PPWC sup. | 2.85 | 1.52 | 2.33 | 3.54 |
| VM+ w/ IO | 4.31 | 0.87 | 1.55 | 7.42 | PPWC syn. | 2.73 | 1.52 | 2.28 | 3.45 |
| VM++ w/o IO | 4.47 | 2.41 | 3.74 | 5.69 | | | | | |
| VM++ w/ IO | 2.26 | **0.75** | **1.16** | 1.90 | | | | | |

Table 3: Quantitative results of Voxelmorph++ (with instance optimisation) on the test cases with different sub-sets of the dataset as training data, reported as mean TRE and 25/50/75% percentiles in mm.

| Setup | TRE | 25% | 50% | 75% |
|---|---|---|---|---|
| initial | 16.25 | 10.14 | 15.94 | 21.76 |
| full dataset | **2.26** | **0.75** | **1.16** | **1.90** |
| TCIA-NSCLC | 5.07 | 1.07 | 2.11 | 8.93 |
| TCIA-NLST | 3.30 | 0.90 | 1.46 | 3.27 |
| Empire10 | 3.35 | 0.90 | 1.47 | 3.57 |
| TCIA-Ventilation | 3.19 | 0.89 | 1.42 | 2.86 |
| L2R-LungCT | 2.90 | 0.88 | 1.41 | 2.63 |

loss to using a combination of MIND features and a Laplace regularisation. The network extends the basic U-Net with a heatmap prediction head. When trained for approx. 6 hours on the proposed dataset, it achieves a TRE of 6.53 mm (feed-forward network only) or 4.31 mm (when followed by test-time instance optimisation (IO)) with a runtime of only 2 s per case.

As a final benchmark method for image-based approaches, we explore the benefits of incorporating the keypoint supervision provided for this new dataset and train the complete **VoxelMorph++** approach [18]. This yields a TRE of 4.47 mm (network only) and 2.26 mm (followed by IO), which is an improvement of nearly 50% solely based on the stronger supervision that is provided.

For point cloud-based registration, we implement the **PointPWC-Net** [56] as it is part of the current SOTA method for point cloud-based lung registration on the PVT dataset [48]. We train the network with two different supervision strategies on the subsampled clouds with 8k points: 1) Training on real inhale-exhale pairs supervised with the automatic correspondences. 2) Similar to [56], training pairs consist of one real cloud and a synthetic deformation (by a 2-scale random field) such that point correspondences are precisely known. The two approaches achieve 2.85 mm and 2.73 mm TRE, respectively, with an inference time of 0.2 s. In addition, we employ the optimisation-based **Coherent Point Drift (CPD)** with implementation choices detailed in the provided source code. Running for 50 iterations in approx. 14 s it reaches 3.13 mm TRE and can be further improved by incorporating the provided vein/artery labels to 2.60 mm (label CPD).

To evaluate how training on the full Lung250M-4B training dataset compares to the use of singular sub-datasets, we performed ablation experiments using Voxelmorph++ with different training setups, using only one of the sub-datasets for each training. Results are presented in Tab. 3 and range from 5.07 mm to 2.90 mm TRE with each setup being outperformed by training on the full dataset. This demonstrates the benefits of our large-scale combined dataset from multiple sources.

All above described experiments are performed with the proposed fixed train/validation/test split. To evaluate the remaining domain gap between training and test data and the suitability of this split, we trained Voxelmorph++ on DIR-LAB COPDgene in a 2-fold cross-validation manner in addition to the full dataset as training data. This setup results in a TRE of 2.10 mm, only slightly better than the achieved 2.26 mm TRE, when training with the proposed split.

**Comparison to Published Work**   One objective of our work is to provide a more comparable benchmark for evaluating 3D deformable (lung) registration on both images and point clouds. Hence, the two previously separated research fields are joined more closely. Current SOTA image-based methods achieved 7.98 mm [2], 3.83 mm [38], 1.34 mm [16] and 0.82 mm [47] on the same test data, where the last method is a non-learning based approach that requires a runtime of 5 minutes. Our best benchmark method yielding 2.24 mm comes close to the current best DL-network. For point cloud SOTA two different preprocessing strategies were proposed in prior work: 1) small but expressive Förstner keypoint clouds that yield a TRE of 2.44 mm with a graph network plus optimiser [15] and 2) large-scale vessel clouds with additional geometric but no semantic features (PVT dataset) that yield a TRE of 2.86 mm using a PointPWC-Net with optimal transport and postprocessing [48]. A PointPWC-Net trained on the PVT1010 dataset under a training setup comparable to the one we employed in our experiments achieves a TRE of 4.50 mm [6].

This highlights that Lung250M-4B dataset is very suitable for both image-based and/or point-cloud-based approaches and hybrid approaches that combine convolutional feature extraction with graph neural networks are likely to benefit. Interestingly, our almost naïve coherent point drift (CPD) baseline can substantially outperform the CPD results in [48] (9.30 mm), which indicates that our data generation that enables both semantic labels and relatively expressive point clouds is useful.

## 6   Technical Limitations

The wide scope of potential methods to extract point clouds from 3D medical volumes and generate labels and correspondences for supervision make it impossible to study the influence of all implementation choices on our provided dataset. The chosen nnUNet architecture has been proven to be very robust for 3D medical segmentation and the 100 manually labelled pulmonary artery trees appear sufficient to train a reliable model. Nevertheless, uncommon anatomical variations or pathologies may impede the quality of point cloud extraction and subsequent analysis of derived measures, e.g. motion vectors and volume change. The provided automatic correspondences are well distributed (see Fig. 3) and yield an excellent agreement with manual landmarks (see Tab. 2) but nevertheless constitute noisy labels. Learning from "perfect" synthetic transformations or at least adding them to a combined supervision may further improve performance (see results of PointPWC in Tab. 2).

The size of our dataset is about **3-4**× larger than previous medical registration benchmarks (apart from inter-subject brain databases) but still considered small in comparison to computer vision. Please note that nearly no other 3D+time datasets with deformable motion and (pseudo) ground truth motion exists otherwise and dynamic FAUST [8] comprises only 10 different subjects. Finally, the set of benchmark methods is not complete, but we strongly believe the provision of Lung250M-4B will encourage researchers from both 3D medical imaging and geometric deep learning (for motion) to challenge their latest methods and to set new best scores.

## 7   Conclusion

We introduced the Lung250M-4B dataset as a public benchmark for large motion exhale-to-inhale lung registration. The dataset stands out from existing benchmarks in three aspects. 1) It is the first combined benchmark for point- and image-based registration. 2) It features 124 images, which is roughly one order of magnitude more than previous large-motion lung datasets. 3) It provides accurate keypoint correspondences for supervised learning.

In our experiments, we evaluated diverse registration methods, including classical and deep learning-based approaches for point- and image-based registration, on the dataset to provide strong baselines. The experiments revealed several insights, pointing to promising future research directions. 1) Classical optimisation methods achieved the best results for both image- and point-based registration, generally demonstrating room for improvement of learning-based methods. 2) DL-based point cloud registration is comparable with DL-based image registration even though point-based methods do not employ IO. Point-based IO is to date barely studied in the literature and could potentially make point-based methods surpass image-based DL methods. 3) The inclusion of artery and vein labels could improve performance of a classical optimization approach (CPD). Hence, they could also be a valuable source of supervision for future training of DL-based methods.

Overall, our work highlights great potential of point cloud-based lung registration. However, to fully understand the potential of point-based methods in medical registration, similar analyses and benchmarks for other anatomies are needed in future work. If our findings transfer to such other anatomies, point-based registration, with the inherent benefits of anonymity preservation, robustness, and computational efficiency, might become an increasingly important component in medical registration.

### Acknowledgments

We are grateful to Rohit Jena and Kayhan Batmanghelich for providing us with a pre-trained version of their vessel segmentation model [26]. We also thank Lasse Hansen for sharing his efficient coherent point drift implementation.

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
