# Supplementary Material: Lung250M-4B

**Fenja Falta**
Institute of Medical Informatics
University of Lübeck
fenja.falta@student.uni-luebeck.de

**Christoph Großbröhmer**
Institute of Medical Informatics
University of Lübeck
c.grossbroehmer@uni-luebeck.de

**Alessa Hering**
Departments of Imaging
Radboud University Medical Center, Nijmegen

**Alexander Bigalke**
Institute of Medical Informatics
University of Lübeck

**Mattias P. Heinrich**
Institute of Medical Informatics
University of Lübeck

## 1   Dataset Retrieval

The download link for the data as well as all code (including preprocessing scripts and benchmark code) and trained models can be retrieved from our GitHub repository: `https://github.com/multimodallearning/Lung250M-4B`

Additionally, the point cloud data is available via the following link: `https://zenodo.org/records/10046885`

The dataset can be set up using the following procedure

1. Download dataset provided by us with 204 scans or 102 patients (including TCIA-NLST, TCIA-Ventilation, L2R-LungCT and TCIA-NSCLC)

2. Consent to usage policy and download DIR-LAB COPDgene dataset (20 scans of 10 patients) from their website `https://med.emory.edu/departments/radiation-oncology/`

    2.1. Preprocess the COPDgene dataset using our preprocessing script from GitHub

3. Consent to usage policy and download EMPIRE10 dataset from grand-challenge `https://empire10.grand-challenge.org`

    3.1. Preprocess EMPIRE10 dataset (24 scans of 12 patients) with our provided script

## 2   Dataset Curation Details

The goal of the Lung250M-4B dataset is to provide a basis for effective research on deformable lung registration using CT scans and point clouds, especially with a focus on learning-based methods. Therefore, for curation, we evaluated potential datasets mainly with 3 criteria: 1) Sufficient motion between scans, 2) variability with respect to subjects, pathologies, and acquisition modalities, and 3) size. Other constraints include free availability of data and sufficient image quality in terms of field of view, resolution, and artefacts. The presented dataset is the result of a trade-off between these aspects. We proceeded as follows: First, we searched for potential datasets and relevant publications in medical databases (TCIA, Pubmed), dataset hosts (Zenodo), and machine-learning challenge platforms (Grand Challenge, Kaggle). We screened potential datasets and discarded all candidates that had low image quality (e.g., cone beam CTs) or insufficient lung motion (e.g., RIDER Lung

dataset [27]). For datasets where there is no external information on motion, we automatically generated lung masks (see Section 4.1 in the main paper) and calculated the volume change between scans of a pair.

The final choice included 6 datasets (COPDgene, EMPIRE10, L2R-LungCT, TCIA-NSCLC, TCIA-Ventialation, TCIA-NLST) with 3 different acquisition modalities (inhale/exhale breath-hold CTs, 4DCTs, respiratory phase-unspecified breath-hold CTs). For our purposes, inhale/exhale breath-hold CTs are most suitable, but these datasets are rare. 4DCTs typically provide images of 10 different respiratory phases but are often artefact-prone. The longitudinal breath-hold images used in the National Lung Trial are numerous, but have little motion between baseline and follow-up scans. While COPDgene, EMPIRE10, and NLST consist at least in part of data derived from lung screenings and may include healthy subjects, TCIA-Ventilation and TCIA-NSLCS patients are diagnosed with lung diseases. Due to the initial anonymisation, it is not possible to provide accurate statistics on the composition of the curated dataset in terms of demographic features. However, due to the high variability of subjects compared to previous datasets, we hope to achieve greater generalisability for potential lung registration methods.

In the following, the source datasets are described in more detail, while an overview can be found in Tab. 1 and Fig. 1.

## 2.1 DIR-LAB COPDgene

The COPDgene dataset consists of data from the COPDGene study, which investigates the influence of genetic factors on the development of COPD in smokers. The trial has been approved by the Institutional Review Boards (IRB) of the individual screening sites. To determine the severity of the disease, CT scans of under normal exhalation and maximum effort inspiration breath hold were acquired in addition to lung function testing. From the pool, 10 scan pairs were randomly selected and manually annotated with corresponding landmarks (n>= 447) at vessel and bronchial bifurcations, from which a final subset of n=300 landmarks has been sampled uniformly. The exact procedure is described in detail in [4]. Due to the large number of landmarks and the extent and complexity of the deformations, the COPDgene dataset is very suitable for evaluation of deformable lung registration. The dataset is not published under a standard licence, but can be downloaded from the project website after filling out a request[1]. Publications using these data must reference [3]. All CT scans have an axial resolution of $512 \times 512$ pixels with a uniform spacing of at least 0.59 and at most 0.742 mm. Each volume consists of 112 to 135 slices with a respective slice thickness of 2.5 mm. We include all scans and annotations in our dataset. While this subset is in principle also applicable for training, we use it as a test set due to the small number of scans and its excellent suitability for evaluation.

## 2.2 Grand Challenge EMPIRE10

The EMPIRE10 dataset [18] was created as part of MICCAI 2010 to evaluate lung registration solutions and is available on the Grand Challenge competition platform. It aims to test registration accuracy regarding a versatile set of clinical tasks. The 30 cases included in this dataset, therefore, come from a variety of exams, subjects and image processing, namely breath-hold inspiration, breath-hold inspiration and expiration, 4D data, ovine data, contrast-noncontrast and artificially warped scan pairs. For our purposes, we use only the breath-hold inspiration and expiration and 4DCT scans (#1,#7,#8,#13,#14,#16,#17,#18,#20,#21,#23,#28) and discard the rest. The former obtains its scans from the Dutch-Belgian randomized lung cancer screening trial (NELSON [26]), which was conducted in a total of 4 medical centres. Subjects of the study were current and former, mostly male heavy smokers aged 50-75 years [28]. The trial was approved by the ethics board of each site and the relevant minister of health [24]. Images were obtained using a low-dose (inhale) and ultra-low-dose (exhale) protocol in a Philips Brilliance 16P and have a pixel spacing of 0.63-0.70 mm and a slice thickness of 1 mm. We include all 8 cases in our dataset. The 4DCT data were obtained using a GE Discovery ST multi-slice PET/CT and Philips Brilliance CT 16 and [18] retrospectively retrieved from the hospital information system. Pixel spacing was 0.98 mm for these scans, while slice spacing varied between 1.25 and 2.50 mm. We included all 4 pairs of data in the Lung250M-4B dataset. EMPIRE10 is not published under any standard license. However, the dataset can be freely

---

[1]`https://med.emory.edu/departments/radiation-oncology/research-laboratories/`
`deformable-image-registration/downloads-and-reference-data/copdgene.html`

downloaded without registration from the challenge website and has to be cited with the original publication[2]. We are providing our pipeline for reprocessing into the Lung250M-4B format.

## 2.3 Grand Challenge Learn2Reg LungCT

Another dataset that exists to benchmark lung registrations on Grand Challenge is L2R-LungCT, which is a subset of the larger continuous Learn2Reg Challenge that addresses other medical image registration tasks, such as registration of multimodal abdominal CT&MR or MRI brain scans [12]. The LungCT dataset includes 30 scan pairs (20 training and 10 test), all of which can be freely obtained under the *CC-BY-4.0* licence[3]. In addition, manual landmarks are available for 3 validation cases. All images were acquired between 2016 and 2017 at Radboud University Medical Center, Nijmegen, NL, and were retrospectively retrieved from the hospital information system. Scan pairs were selected according to three criteria: Only (I) breathhold scans that (II) had sufficient lung coverage in both images and (III) had at least 300 slices were included. All images have an axial resolution of 512 by 512 pixels and a uniform spacing between 0.56 and 0.81 mm and between 321 and 705 slices with a slice thickness of 0.5 mm. Use of data from Radboud University Medical Center was approved by the institutional review board under an umbrella protocol for "Retrospective research reusing care data within the Department of Radiology and Nuclear Medicine". This approved document grants access to retrospective and anonymised imaging data for research purposes in Radboud UMC. We chose to include the whole 30 scan-pairs in Lung250M-4B.

## 2.4 TCIA Ventilation

The TCIA-Ventialtion data collection[7, 5] includes image data from a study evaluating the accuracy of pulmonary ventilation measurements in breath-hold CT, 4DCT, and Galligas PET examinations conducted at the Royal North Shore Hospital, Sydney between 2013 and 2015 [6]. The trial is approved by the local health district ethics committee and registered with the Australian New Zealand Clinical Trials Registry (ACTRN12612000775819) and the collection is available through TCIA with licence *CC-BY-4.0*[4]. We include all (20) breath hold scan pairs in our dataset. All acquisitions were performed with a Siemens Biograph mCT.S/64 PET/CT scanner in combination with audiovisual feedback for ten-second breath hold control at approximately 80% maximal inspiration and expiration, respectively. A large proportion of patients (at least 16) possessed pathological impairments with COPD and lung tumours. All scans have a resolution of 512 by 512 voxels and between 153-193 slices and a uniform spacing of $0.97 \times 0.97 \times 2$ mm.

## 2.5 TCIA NSCLC

Furthermore, we include 20 scan pairs from the TCIA-NSLCS data collection[13, 14, 2, 21, 5]. The dataset includes scans performed between 2008 and 2012 at VCU Massey Cancer Center in the Department of Radiation Oncology, VA, USA on a total of 20 patients with non-small cell lung cancer undergoing image-guided radiotherapy. Patients consented to participate in the prospective study, which was approved by the institutional review board. In total, the collection includes 82 4DCTs and 507 4D Cone Beam CTs images acquired before and during radiotherapy. Audiovisual feedback was used for all scans to minimise respiratory irregularities. Tumours were located at various locations in the lung and occupied a mean volume of 76 cm2. Limitations of these collections are 4DCT sorting artefacts, as mentioned in the original dataset publication. The dataset was published under the *CC-BY-3.0* license on TCIA[5]. From this collection, we select the scans with maximum inhalation and exhalation from a total of 10 respiratory phases from the pre-therapeutic 4DCT images. All acquired images share the same axial resolution and spacing of $512 \times 512$ pixels and 0.97 mm respectively.

## 2.6 TCIA NLST

The National Lung Screening Trial (NLST) was conducted in the United States between 2002 and 2004 to compare the efficacy of chest radiography and low-dose CT for the early detection of lung

---

[2] https://empire10.grand-challenge.org/Download/
[3] https://zenodo.org/record/4279348
[4] https://wiki.cancerimagingarchive.net/pages/viewpage.action?pageId=125600096
[5] https://wiki.cancerimagingarchive.net/pages/viewpage.action?pageId=21267414

tumours in high-risk individuals (heavy smokers aged 55 to 74 years)[23]. These individuals were offered 3 screenings at 1-year intervals in both arms of the study. Before randomisation, all subjects completed an informed consent form developed and approved by the institutional review boards of the screening centres and the National Cancer Institute.

Approximately 73,000 low-dose CT scans from the trial are available through TCIA[6] under the *CC-BY-4.0* licence [20, 5]. From this large set, we randomly extracted 281 scan pairs, with baseline and follow-up included in a span of one year. Because the NLST study protocol does not require inhale/exhale breath-hold examinations, a large proportion of the scans do not have sufficient lung motion for our purposes. We identified candidate pairs by predicting and comparing lung masks for every pair of scans. We excluded all patients with a lung volume change of less than 380 ml, resulting in a total of 22 scan pairs. Because of anonymisation, individual scanners and screening sites can no longer be attributed. But, it is likely that cases from different scanners are selected.

For validation within this sub-dataset, we used an additional 10 NLST scan pairs, for which landmark annotations were published through the Learn2Reg challenge [12]. In baseline and follow-up scan, a total of 100 corresponding landmark pairs were identified and made available under the CC-BY-4.0 licence. Since the selection of these scan pairs is not subject to the above criteria, they have a lower lung volume change (mean 247ml) in comparison. The acquired 32 scan pairs have an inplane axial resolution of 512 by 512 voxels with a spacing between 0.47 and 0.9 mm with a slice thickness between 1 and 3.2 mm.

---

[6]https://wiki.cancerimagingarchive.net/display/NLST/National+Lung+Screening+Trial

Table 1: Overview about the Composition of Lung250M-4B. Landmark Cases denoted with * have been created within the scope of this work. Abbreviations: BH(-I/-E) = Breath-Hold(-Inspiration/-Expiration), IRB = Institutional Review Board, FOV = Field of View)

| | Study Focus | Subject Information | Subject Pathologies | Acquisition Centers (Location) | IRB Study Approvement | Selected Examination Types | Release Focus | Licence | IRB Release Approvement | Scanner Information | #Selected Scan Pairs | #Landmark Cases | FOV | Original Pixel Spacing | Original Slice Thickness |
|---|---|---|---|---|---|---|---|---|---|---|---|---|---|---|---|
| DIR-LAB COPDgene | Genetic factors for the development of COPD in smokers | Male/Female non-Hispanic whites or African-American aged 45-80 years with >= 10 pack-year smoking history | COPD (severity undisclosed), other pulmonary diseased excluded from study | 21 (USA) | ✓ | BH-I / BH-E | Benchmark | Ambiguous | N/A | GE Lightspeed VCT | 10 | 10 | Full Lung / Full Lung | 0.590 - 0.652 mm | 2.5 mm |
| Empire10 (Data Composition) | BH: Lung Cancer Screening (Nelson Study CITE) 4DCT: Undisclosed | BH: Mostly Males from NL and BE, aged 50-75, >= 15 cigarettes daily for 25 years or 10 cigarettes daily for 30 years 4DCT: Undisclosed | N/A | 4+ (NL,BE) | BH: ✓ 4DCT: N/A | BH-I / BH-E 4DCT | Challenge | Ambiguous | N/A | Various | 12 | 2* | Full Lung / Full Lung | 0.7-2.5 mm | 1-2.5 mm |
| L2R-LungCT | Retrospective Medical Data | Undisclosed | Undisclosed | 1 (NL) | N/A | BH-I / BH-E | Challenge | CC-BY-4.0 | (✓) | TOSHIBA Aquilion | 30 | 3 | Full Lung / Partial Lung | 0.56-0.8 mm | 0.5 mm |
| TCIA-Ventilation | Ventilation Measurement Evaluation | Male/Female Lung Cancer Patients aged 54-73 | COPD (mild to severe), Lung tumour | 1 (AU) | ✓ | BH-I / BH-E | Medical Study | CC-BY-4.0 | Implicitly though TCIA TOS | Siemens Biograph mCT.S/64 PET/CT | 20 | 2* | Full Lung / Full Lung | 0.97 mm | 2 mm |
| TCIA-NSCLC | Image-Guided Radiotherapy | Undisclosed | Non-Small Lung Cancer (Stages IIA-IIIB) | 1 (USA) | ✓ | 4DCT | Medical Study | CC-BY-3.0 | Implicitly though TCIA TOS | Philips Brilliance Big Bore | 20 | - | Full Lung / Full Lung | 0.97 mm | 3 mm |
| TCIA-NLST | Efficacy of Chest X-Ray and Low-Dose CT in Lung Cancer Screenings | age 55-74, >= 3B pack-years of cigarette smoking | N/A | 33 (USA) | ✓ | Low-Dose BH-CT | Medical Study | CC-BY-4.0 | Implicitly though TCIA TOS | Various | 32 | 10 | Full Lung / Full Lung | 0.47-0.9 mm | 1-3.2 mm |

## 2.7 Additional Manual Landmarks

For some of the datasets (NLST, L2R-Lung, COPDgene) manual landmarks could be adopted for validation and testing. For the EMPIRE10 (cases #8 and #20) and TCIA-Ventilation (cases #11 and #14) sub-datasets, additional landmark annotations were created as part of this work. The readers (1 researcher and 2 research assistants with medical background) matched 100 corresponding landmarks in each scan pair using the freely available semi-automatic annotation software *isimatch*[7] [17], following the procedure in [18]. For each inhale scan, a lung mask was created using an nnUNet [15]. Then, 100 points destined by the *Distinctive Point Finder* module were automatically selected based on the image gradients of their surroundings within the lung mask, which are usually located at visible vessels and bronchial bifurcations. Subsequently, the readers identified these corresponding points in the exhale scans. Based on a thin-plate-spline interpolation, the software allows automatic matching of the points after a sufficient number of manual identifications. All automatic landmarks were manually reviewed and corrected if necessary. We further estimated the inter-observer error of both annotators using *isimatch* in 4 registration pairs for a total of 400 landmarks and obtain a mean error and standard deviation of 0.79 mm and 1.24 mm respectively, which compares well to the isotropic image resolution of 1 mm.

## 3 Licence

We publish all data under CC-BY-4.0 licence.

For all image data that we may not redistribute due to their licencing (i.e. COPDgene and EMPIRE10), we include detailed instructions on how to obtain the data and provide preprocessing scripts in our GitHub repository.

This dataset is intended for research purposes only and not for clinical usage.

## 4 Dataset Structure

The dataset is divided into multiple types of instances with the following folder structure:

- **imagesTr/imagesTs**: Preprocessed CT scans as .nii.gz files
- **masksTr/masksTs**: Lung masks as .nii.gz files
- **segTr/segTs**: Vessel segmentations as .nii.gz files
- **cloudsTr/cloudsTs**: Point clouds and features, each in a list containing 1.) the point cloud sampled to 8196 points, 2.) the point cloud of the vessel skeleton and 3.) the full point cloud
    - **coordinates**: (x,y,z) coordinates of each point
    - **artery_vein**: label of each point (1: vein, 2: artery)
    - **distance**: distance from each point to the closest vessel edge
- **corrfieldTr**: CorrField keypoint correspondences

Folders ending in Tr contain files for training, folders ending in Ts contain files for validation and test. All files have a 3-digit case identifier (000 to 123) in their name. Additionally whether the file corresponds to the in- or expiratory phase is indicated by a 1 or 2 respectively at the end of the file name.

## 5 Data Samples and Statistics

A visualisation of image and point cloud data from each data subset can be seen in Fig. 1. Statistics on the volume of the segmentations and the sizes of the point clouds can be seen in Tab. 2 and Fig. 2. Each skeleton cloud contains more than 8196 points and can thus be downsampled to obtain all three types of point clouds.

---

[7]https://www.isi.uu.nl/research/software/isimatch/

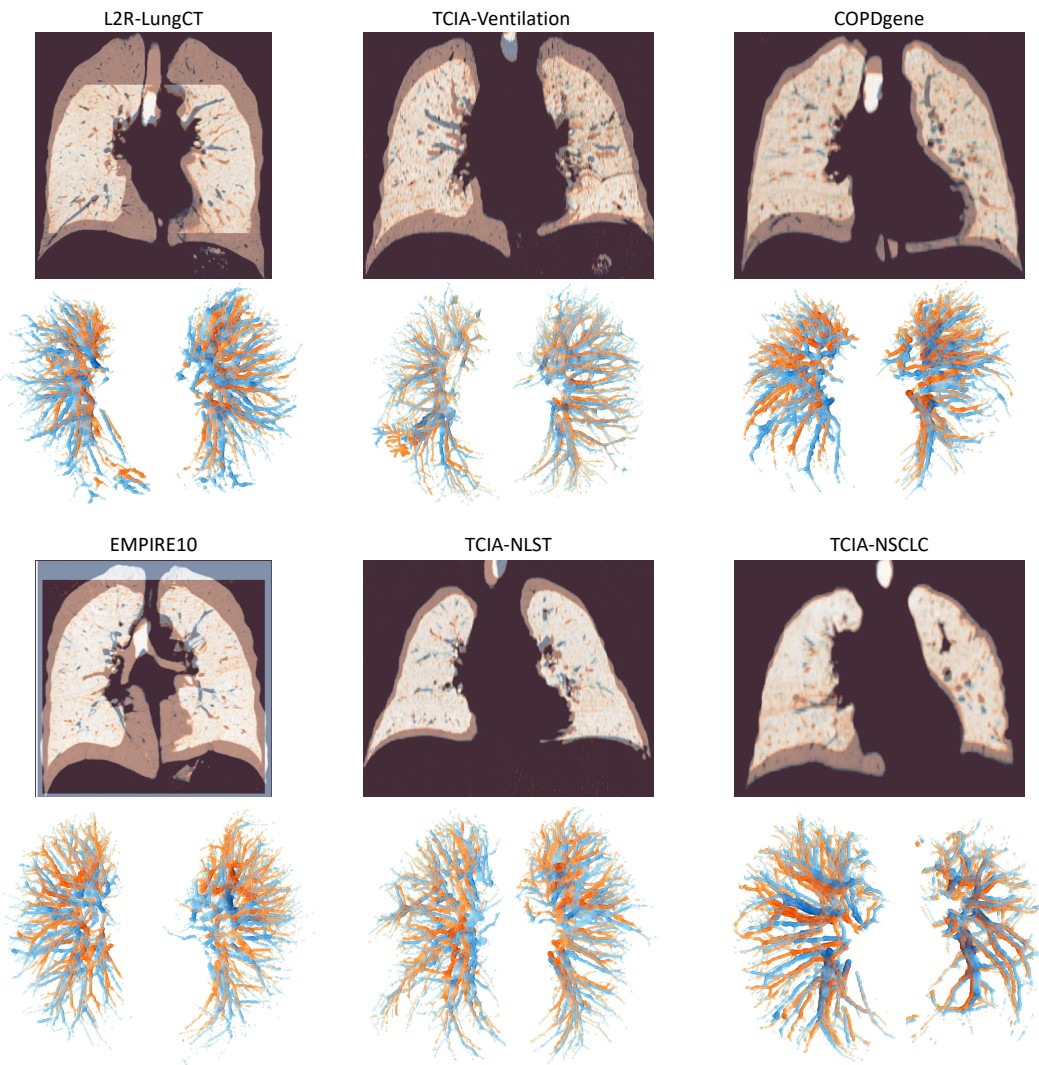

Figure 1: Visualisation of six sample cases from our dataset, including sample from each subset. For each case, we show an overlay of fixed (blue) and moving (orange) CT slices (top) and skeletonised 3D lung vessel trees (bottom).

## 6 Benchmark Methods

For implementation details regarding the benchmark methods, we refer to our GitHub.

### 6.1 corrField

We apply corrField [11], which requires no training and is provided with GPU acceleration in our repository, with default parameters as described on the algorithm page `https://grand-challenge.org/algorithms/corrfield`. That means the fixed and moving CT scans alongside a lung mask for the fixed image are provided as input. The number of Föerstner 3D keypoints, for which correspondences are computed in the two-stage discrete optimisation, varies between 5'000 and 8'000.

### 6.2 deeds

Another optimisation-based baseline is deeds [10], which is also run without modifications to the code provided at `https://github.com/mattiaspaul/deedsBCV`. However, to improve the alignment

Table 2: Mean lung and vessel volumes (and their standard deviation) for each source dataset. Lung volumes are stated in ml, point cloud sizes in number of points.

| Source Dataset | Lung Volume (Insp. Phase) | Lung Volume (Exp. Phase) | Size of Point Cloud | Size of Skeleton Cloud |
|---|---|---|---|---|
| DIR-LAB COPDgene | $4960 \pm 1139$ | $3190 \pm 798$ | $967k \pm 196k$ | $25k \pm 7k$ |
| Empire10 | $6214 \pm 1940$ | $4263 \pm 1207$ | $1249k \pm 193k$ | $31k \pm 10k$ |
| L2R-LungCT | $4841 \pm 1095$ | $2685 \pm 544$ | $1046k \pm 330k$ | $28k \pm 12k$ |
| TCIA-NSCLC | $3887 \pm 1287$ | $3460 \pm 1218$ | $805k \pm 242k$ | $18k \pm 5k$ |
| TCIA-Ventilation | $4666 \pm 969$ | $3528 \pm 1021$ | $1236k \pm 343k$ | $31k \pm 9k$ |
| TCIA-NLST | $6224 \pm 1692$ | $5361 \pm 1419$ | $1385k \pm 337k$ | $40k \pm 12k$ |

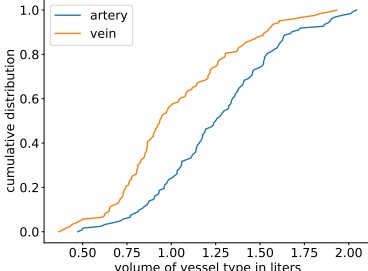 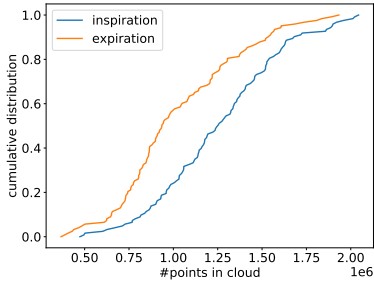

Figure 2: Cumulative distribution of artery and vein volume (left) and number of points in the in- and expiratory point clouds (right).

of inner-lung structures we mask out the background outside of the provided lung segmentations. Since deeds has so far only been parallelised on CPU the run times are substantially higher (minutes rather than seconds). We are sharing a custom script that converts the output displacements into corresponding CSV-files.

### 6.3 VoxelMorph++

We use VoxelMorph++ [9] that substantially extends upon the popular baseline [1] and adapt the implementation of `https://github.com/mattiaspaul/VoxelMorphPlusPlus`. The original code had some shortcomings, namely the inability to work with differently shaped 3D volumes across scan pairs. We implement appropriate padding and cropping operations to fulfil the requirements of input dimensions divisible by 32 for the underlying U-Net. Two variants with a convolutional heatmap regression head, **supervised** and **unsupervised** are trained for 400 and 800 epochs respectively with an initial learning rate of $\eta = 0.001$ and a step reduction of 0.5 after 30 epochs and restarts after every 200 epochs (mini-batch size is one). The loss for **supervised** uses the Euclidean distance with respect to the corrField correspondences (i.e. the target registration error) as described above, whereas **unsupervised** uses a combined image-metric (MIND-loss with MSE) and a Laplacian graph regularisation with a weighting of $\lambda = 0.25$. Further details are provided in our repository and the paper. Both methods use affine augmentation with a strength of the Gaussian random transform of 0.035 (alternating for fixed and moving images). We evaluate the trained models either with or without instance optimisation. The former optimises a combined MIND-metric and diffusion regularisation ($\lambda = .65$) on a dense grid with control point spacing of 2 voxels for 50 iterations using Adam optimiser ($\eta = 1$).

### 6.4 PointPWC-Net

We use the default architecture of PointPWC-Net from [25] without BatchNorm layers and the multi-scale loss from [25] as the objective function. Network parameters are optimised with the Adam optimiser for 1500 epochs (=36k iterations) with a batch size of 4. The initial learning is set to

0.001 and decreased by a factor of 10 after 1200 and 1400 epochs. The network is trained with the following two supervision strategies.

**Supervised learning with corrField correspondences**   The corrField algorithm provides a set of keypoint correspondences for each data pair. We interpolate the corresponding displacement vectors of the moving keypoints to the moving points in the input cloud to our network and use the resulting flow vectors for direct supervision. Given a pair of point clouds along with this flow vector, we randomly perform one of the following two augmentation strategies at training. 1) We apply a random rigid transformation (scaling, rotation, translation) to either the fixed or moving cloud while the other one remains unaffected. 2) We augment both clouds with the same rigid transformation. In both cases, the underlying flow field is transformed accordingly.

**Learning on synthetic deformation**   The general idea of this strategy is to train the model on pairs consisting of a real cloud and a synthetic deformation of it such that point-wise displacement vectors are precisely known. Similar to [22], we generate deformations with a 2-scale random field through the following steps.

1. For a given real pair of fixed and moving clouds with 16k points each, we randomly sample either the fixed or the moving cloud as the initial cloud $X$ to be deformed.

2. We randomly sample a set of 500 local control points $x_{loc}$ from $X$.

3. For each local control point, we sample a random displacement vector $\Delta^{(loc)} \in \mathbb{R}^3$, with $\Delta_i^{(loc)}$ uniformly drawn from [-3 mm,3 mm].

4. We interpolate the displacements from the local control points to the full cloud $X$ with an isotropic Gaussian kernel ($\sigma = 15$mm) and displace the points accordingly, yielding the locally deformed cloud.

5. To the latter cloud, we apply voxel downsampling with a voxel size of 90 mm to obtain a set of roughly 10-30 global control points $x_{glob}$

6. For each global control point, we sample a random displacement vector $\Delta^{(glob)} \in \mathbb{R}^3$, with $\Delta_i^{(glob)}$ uniformly drawn from [-25 mm,25 mm]

7. We interpolate the displacements from the global control points to the full locally deformed cloud with an isotropic Gaussian kernel ($\sigma = 25$mm) and displace the points accordingly, yielding a locally and globally deformed cloud $X_{def}$.

8. Since all the previous operations preserve point correspondences, displacement vector fields for supervision are precisely known, leaving us with the pair $(X, X_{def})$ as input and $X_{def} - X$ as the corresponding ground truth.

9. As is, $X$ and $X_{def}$ exhibit precise point correspondences, which is not realistic and might cause overfitting. Therefore, we sample two disjoint subsets of 8k points from $X$ and $X_{def}$ as the final input to our network and keep the displacement vectors corresponding to the points in the moving cloud for supervision.

## 6.5   Coherent Point Drift

Finally, we explore Coherent Point Drift (CPD) a classical untrained deformable point cloud registration method [19]. Following [8] we set the hyperparameters to $\omega = 0.5$, $\epsilon = 10^{-5}$, $\lambda = 8$ and $\beta = 1.25$ and optimise for 50 iterations. CPD models both point clouds as multivariate Gaussian mixture models and alternates between the fitting of a transformation and the point distributions in an expectation-maximisation algorithm. We extend the method by leveraging the automatic anatomical labels that assign either vein or artery to each point in the cloud. Consequently, we introduce another weight $\alpha = 0.05$ and set $\beta = 1$ to balance the influence of these new features. $\alpha$ was fine-tuned on a single validation case. Our GPU implementation can be found in the GitHub repository.

## 6.6   Results

The performance of the above methods on the 10 test cases from the COPDgene dataset was already reported in Tab. 2 of the main paper. Here, we evaluate the same methods on the 17 validation cases

Table 3: Quantitative results on the 17 validation cases of image-based (left) and point-based (right) methods, reported as mean TRE and 25/50/75% percentiles in mm. IO: Instance optimisation

| Method | TRE | 25% | 50% | 75% | Method | TRE | 25% | 50% | 75% |
|--------|------|------|------|------|--------|------|------|------|------|
| initial | 14.02 | 8.18 | 12.21 | 18.07 | initial | 14.02 | 8.18 | 12.21 | 18.07 |
| corrField | **2.14** | **1.08** | **1.66** | **2.42** | CPD | 4.21 | 1.60 | 2.46 | 3.85 |
| deeds | 2.26 | 1.14 | 1.75 | 2.59 | CPD w/ labels | 3.90 | **1.49** | **2.30** | **3.52** |
| VM+ w/o IO | 5.50 | 2.76 | 4.40 | 7.14 | PPWC sup. | **3.12** | 1.58 | 2.45 | 3.74 |
| VM+ w/ IO | 3.69 | 1.23 | 2.05 | 3.91 | PPWC syn. | 3.29 | 1.67 | 2.54 | 3.94 |
| VM++ w/o IO | 4.20 | 2.33 | 3.52 | 5.29 | | | | | |
| VM++ w/ IO | 2.67 | 1.15 | 1.85 | 2.89 | | | | | |

Table 4: Quantitative results of Voxelmorph++ (with instance optimisation) on the validation cases with different sub-sets of the dataset as training data, reported as mean TRE and 25/50/75% percentiles in mm.

| Setup | TRE | 25% | 50% | 75% |
|-------|------|------|------|------|
| initial | 14.02 | 8.18 | 12.21 | 18.07 |
| full dataset | **2.67** | **1.15** | 1.85 | 2.89 |
| TCIA-NSCLC | 4.07 | 1.23 | 2.05 | 4.28 |
| TCIA-NLST | 3.04 | **1.15** | **1.84** | **2.85** |
| Empire10 | 3.69 | 1.21 | 1.94 | 3.25 |
| TCIA-Ventilation | 3.27 | 1.17 | 1.89 | 3.07 |
| L2R-LungCT | 3.32 | 1.22 | 1.95 | 3.26 |

for which we provide manually annotated landmark correspondences. Results are shown in Tab. 3 and Tab. 4 and reveal the following findings. First, classical image-based methods perform worse than on the test cases but still achieve the top performance among all methods. Second, learning-based methods for image registration achieve slightly improved results (apart from VM++ w/ IO). Third, for point-based registration, both versions of CPD deteriorate and are now, at least on average, inferior to both versions of the learning-based PPWC, which we primarily attribute to a particularly challenging case, where CPD completely failed. Fourth, as on the test set, PPWC achieves competitive scores with image-based DL methods, being only inferior to VM++ with IO. Fifth, consistent with the test results, training on the full dataset performs better than training on sub-datasets. While training on only L2R-LungCT achieved the best results on the test dataset, TCIA-NLST here achieves the lowest TRE, which is likely due to the larger number of TCIA-NLST validation cases compared to other sub-datasets. Finally, we visualise qualitative results for image and point cloud registration in Figs. 3 and 4, demonstrating largely accurate and smooth alignments of the lungs.

# 7 Datasheet

## 7.1 Motivation

- **For what purpose was the dataset created?** Was there a specific task in mind? Was there a specific gap that needed to be filled? Please provide a description.
  **A:** Lung250M-2B was created as a dataset to train and evaluate methods for deformable 3D registration on images, point-clouds or both. Compared to other lung registration datasets, we a) provide a large number of scan pairs with large deformations between scans and b) provide 3D point clouds for each scan to evaluate point cloud-based methods **on the same instances** in unison with image-based ones. Up to now, there was a scientific gap for a dataset with large-scale deformable 3D motion and supervision for vision research. Kitti [16] provides mainly part-wise rigid motion, whereas PVT1010 [22] contains similarly expressive deformable point-clouds but without supervision. Our aim is to stimulate research that bridges the methodological limitations of either image-based or point-based 3D registration and e.g. uses features derived from on modality to inform the other.

case #008 (EMPIRE10)   case #094 (TCIA-Ventialtion)   case #114 (NLST)

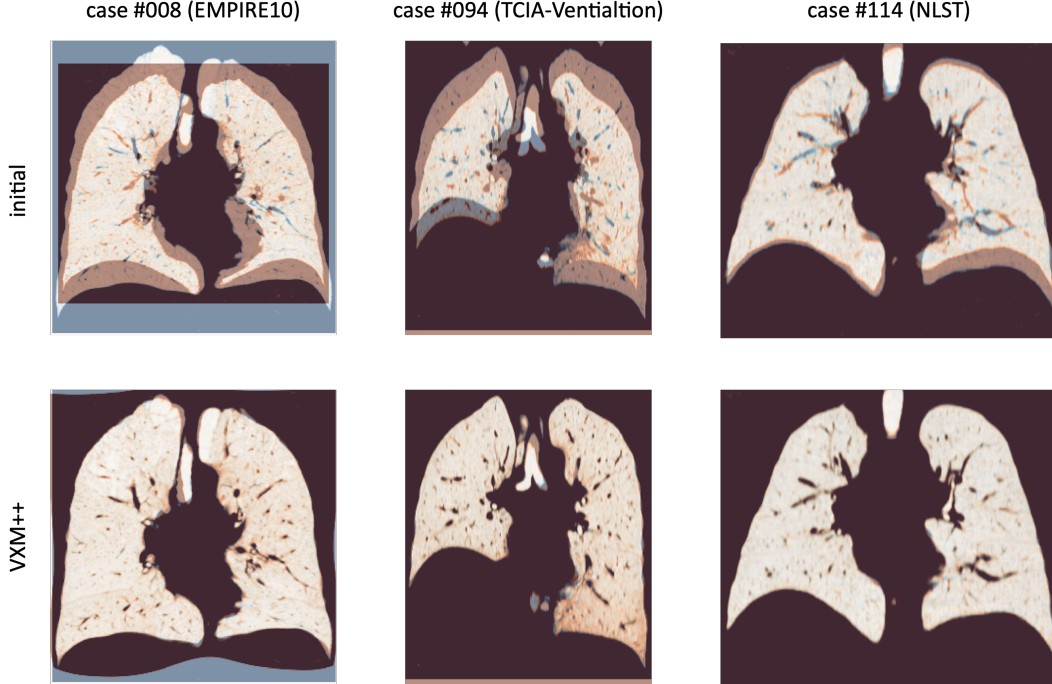

Figure 3: Qualitative results on three sample cases from the validation set. We display initial (top row) and Voxelmorph++-registered (bottom row) overlays of fixed (blue) and warped moving (orange) images.

- **Who created the dataset (e.g., which team, research group) and on behalf of which entity (e.g., company, institution, organization)?**
  **A:** The dataset was created by the authors. (University of Lübeck, Germany).

- **Who funded the creation of the dataset?** If there is an associated grant, please provide the name of the grantor and the grant name and number.
  **A:** A small part of the work was funded by a German federal research grant (BMBF) under the ID 01KD2212A for making available datasets with impact on knowledge gain and research in oncological data science.

- **Any other comments?**
  **A:** No.

### 7.2   Composition

- **What do the instances that comprise the dataset represent (e.g., documents, photos, people, countries)?** Are there multiple types of instances (e.g., movies, users, and ratings; people and interactions between them; nodes and edges)? Please provide a description.
  **A:** For each case (corresponding to one patient) there is paired data corresponding to the in- and expiratory phases. We provide the following type of data:
    - images of CT scans
    - vessel segmentations
    - vessel point clouds
    - point features
    - keypoint correspondences
    - landmarks

- **How many instances are there in total (of each type, if appropriate)?**
  **A:** For each type except landmarks, there are 248 instances, 124 for inspiratory phases and 124 for expiratory phases respectively. We provide 54 instances of landmark annotations, 27 for each phase.

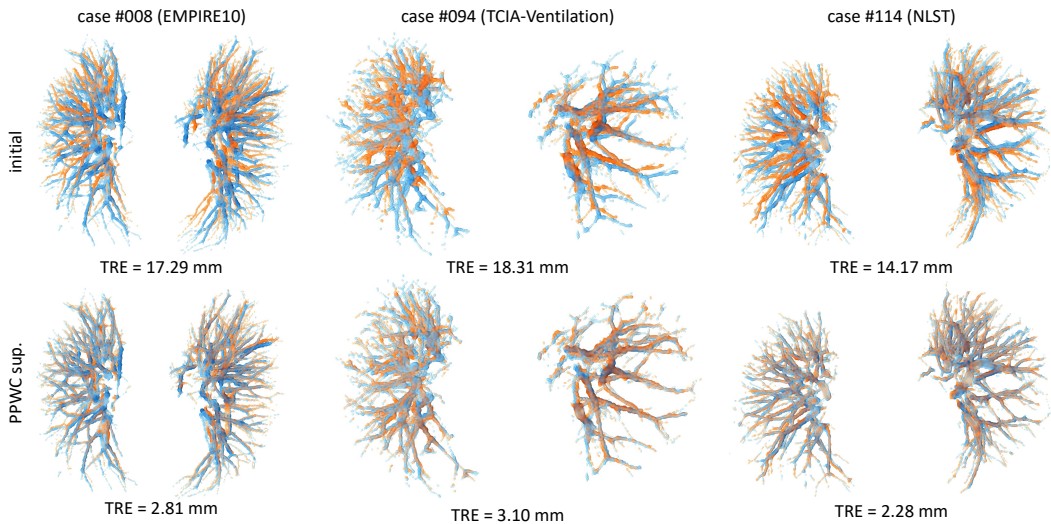

case #008 (EMPIRE10)  case #094 (TCIA-Ventilation)  case #114 (NLST)

initial

TRE = 17.29 mm  TRE = 18.31 mm  TRE = 14.17 mm

PWC sup.

TRE = 2.81 mm  TRE = 3.10 mm  TRE = 2.28 mm

Figure 4: Qualitative results on three sample cases from the validation set. We display overlays of the skeletonised clouds of the fixed (blue) and warped moving (orange) clouds. We calculated the flow on the skeletonised clouds by interpolating the predicted flow on the 8k clouds with an isotropic Gaussian kernel. The first row shows the initial overlap and the second row the registration by the supervised PointPWC-Net.

- **Does the dataset contain all possible instances or is it a sample (not necessarily random) of instances from a larger set?** If the dataset is a sample, then what is the larger set? Is the sample representative of the larger set (e.g., geographic coverage)? If so, please describe how this representativeness was validated/verified. If it is not representative of the larger set, please describe why not (e.g., to cover a more diverse range of instances, because instances were withheld or unavailable).
  **A:** Lung250M-4B is based on image data from other datasets. We selected appropriate cases based on criteria described in Section 2. Due to the variety of original datasets we sampled from, we achieve a high diversity regarding e.g. scanner type, pathologies or examination.

- **What data does each instance consist of?** "Raw" data (e.g., unprocessed text or images) or features? In either case, please provide a description.
  **A:** We provide preprocessed images alongside extracted information (keypoints, landmarks, segmentations) based on these images.

- **Is there a label or target associated with each instance?** If so, please provide a description.
  **A:** For each instance (image pair), we provide weak labels for learning (point features, keypoint correspondences). Additionally, for 27 cases, we provide manual landmarks (usually 100-300 pairs) to evaluate registration accuracy.

- **Is any information missing from individual instances?** If so, please provide a description, explaining why this information is missing (e.g., because it was unavailable). This does not include intentionally removed information, but might include, e.g., redacted text.
  **A:** No.

- **Are relationships between individual instances made explicit (e.g., users' movie ratings, social network links)?** If so, please describe how these relationships are made explicit.
  **A:** All data are enumerated with a case number and a denominator on whether they relate to the in- or expiratory phase.

- **Are there recommended data splits (e.g., training, development/validation, testing)?** If so, please provide a description of these splits, explaining the rationale behind them.
  **A:** We suggest a training/validation/test split. Training data includes all data without landmark annotations, test data are made up of all DIR-LAB COPDgene cases and validation data are made up of all additional cases we provide landmarks for. This split is made clear in the data structure.

- **Are there any errors, sources of noise, or redundancies in the dataset?** If so, please provide a description.
  **A:** Occurring noise in our image data is equivalent to the noise in the original image data. Since we automatically predicted vessel segmentations using the nnUNet framework, segmentations may naturally contain false positives or negatives.

- **Is the dataset self-contained, or does it link to or otherwise rely on external resources (e.g., websites, tweets, other datasets)?** If it links to or relies on external resources, a) are there guarantees that they will exist, and remain constant, over time; b) are there official archival versions of the complete dataset (i.e., including the external resources as they existed at the time the dataset was created); c) are there any restrictions (e.g., licenses, fees) associated with any of the external resources that might apply to a dataset consumer? Please provide descriptions of all external resources and any restrictions associated with them, as well as links or other access points, as appropriate.
  **A:** The DIR-LAB COPDgene and EMPIRE 10 datasets have no CC licence, so we refer to their official website to obtain the data and provide a preprocessing script to generate the preprocessed images. The remaining dataset is self-contained.

- **Does the dataset contain data that might be considered confidential (e.g., data that is protected by legal privilege or by doctor–patient confidentiality, data that includes the content of individuals' non-public communications)?** If so, please provide a description.
  **A:** No.

- **Does the dataset contain data that, if viewed directly, might be offensive, insulting, threatening, or might otherwise cause anxiety?** If so, please describe why.
  **A:** No.

If the dataset does not relate to people, you may skip the remaining questions in this section.

- **Does the dataset identify any subpopulations (e.g., by age, gender)?** If so, please describe how these subpopulations are identified and provide a description of their respective distributions within the dataset.
  **A:** We do not publish explicit metadata on demographic features. However, when publishing high-resolution medical image data, there is necessarily the possibility of deriving such information from the images with a certain degree of probability. Nevertheless, we do not see any additional risk with this publication, as the data is already freely available on the Internet.

- It is possible to indirectly infer meta-information about the patient from the image data. We however do not explicitly provide this information in our dataset.

- **Is it possible to identify individuals (i.e., one or more natural persons), either directly or indirectly (i.e., in combination with other data) from the dataset?** If so, please describe how.
  **A:** Clearly defined de-identifying measures exist for a portion of the dataset (those distributed via TCIA). The other datasets are also anonymized, so reidentification is not possible from our point of view. We take up this point in the main submission under Section 3 (Ethical Discussion). We do not see any additional risk with this publication, as the data is already freely available.

- **Does the dataset contain data that might be considered sensitive in any way (e.g., data that reveals race or ethnic origins, sexual orientations, religious beliefs, political opinions or union memberships, or locations; financial or health data; biometric or genetic data; forms of government identification, such as social security numbers; criminal history)?** If so, please provide a description.
  **A:** The dataset contains health data. However, we do not see any additional risk with this publication, as the data is already freely available.

- **Any other comments?**
  **A:** No.

### 7.3 Collection Process

- **How was the data associated with each instance acquired?** Was the data directly observable (e.g., raw text, movie ratings), reported by subjects (e.g., survey responses), or

indirectly inferred/derived from other data (e.g., part-of-speech tags, model-based guesses for age or language)? If the data was reported by subjects or indirectly inferred/derived from other data, was the data validated/verified? If so, please describe how.
**A:** All data is indirectly inferred from the CT images of the original datasets. We trained and validated several methods on the dataset. The results show that training on the data we derived reduced the target regression error in an expected manner. In our view, this validates the suitability of our dataset.

- **What mechanisms or procedures were used to collect the data (e.g., hardware apparatuses or sensors, manual human curation, software programs, software APIs)?** How were these mechanisms or procedures validated?
  **A:** We attach great importance to reproducibility and the possibility of validation of our methods. Our entire automated workflow is described and deposited in the linked GitHub and can be validated by any expert. The elaborate annotation of landmarks in the lung was performed with peer-reviewed, freely available software and described in detail.

- **If the dataset is a sample from a larger set, what was the sampling strategy (e.g., deterministic, probabilistic with specific sampling probabilities)?**
  **A:** We sampled scan pairs based on total lung volume change and availability of landmark annotations, which is further discussed in Section 2. Point clouds were downsampled based on point cloud density. Details are described in the corresponding section.

- **Who was involved in the data collection process (e.g., students, crowdworkers, contractors) and how were they compensated (e.g., how much were crowdworkers paid)?**
  **A:** The manual annotations were performed by two research assistants (students) employed at the University of Lübeck. They were compensated with a salary of 13€/h, which is the standard salary for this position. We plan to integrate both assistants into further projects.

- **Over what timeframe was the data collected? Does this timeframe match the creation timeframe of the data associated with the instances (e.g., recent crawl of old news articles)?** If not, please describe the timeframe in which the data associated with the instances was created.
  **A:** Scans of the original datasets were acquired between 2002 and 2017. Details are described in Section 2.

- **Were any ethical review processes conducted (e.g., by an institutional review board)?** If so, please provide a description of these review processes, including the outcomes, as well as a link or other access point to any supporting documentation.
  **A:** Not on our end. However, ethical reviews were conducted regarding the data acquisition of the original datasets.

- **Did you collect the data from the individuals in question directly, or obtain it via third parties or other sources (e.g., websites)?**
  **A:** CT scans were sourced from the websites providing the original datasets.

- **Were the individuals in question notified about the data collection?** If so, please describe (or show with screenshots or other information) how notice was provided, and provide a link or other access point to, or otherwise reproduce, the exact language of the notification itself.
  **A:** No.

- **Did the individuals in question consent to the collection and use of their data?** If so, please describe (or show with screenshots or other information) how consent was requested and provided, and provide a link or other access point to, or otherwise reproduce, the exact language to which the individuals consented.
  **A:** We refer to the collection details of the original datasets. All images we use are publicly available.

- **If consent was obtained, were the consenting individuals provided with a mechanism to revoke their consent in the future or for certain uses?** If so, please provide a description, as well as a link or other access point to the mechanism (if appropriate).
  **A:** For this point we refer to the original publications of the data, since we ourselves have not collected a consent.

- **Has an analysis of the potential impact of the dataset and its use on data subjects (e.g., a data protection impact analysis) been conducted?** If so, please provide a description of this analysis, including the outcomes, as well as a link or other access point to any supporting

documentation.
**A:** No.

- **Any other comments?**
  **A:** No.

## 7.4 Preprocessing/cleaning/labeling

- **Was any preprocessing/cleaning/labeling of the data done (e.g., discretization or bucketing, tokenization, part-of-speech tagging, SIFT feature extraction, removal of instances, processing of missing values)?** If so, please provide a description. If not, you may skip the remaining questions in this section.
  **A:** Preprocessing of CT images includes resampling and cropping. Labeled segmentations were obtained via a nnUNet.

- **Was the "raw" data saved in addition to the preprocessed/cleaned/labeled data (e.g., to support unanticipated future uses)?** If so, please provide a link or other access point to the "raw" data.
  **A:** All raw data are publicly available as part of the original datasets.

- **Is the software that was used to preprocess/clean/label the data available?** If so, please provide a link or other access point.
  **A:** Preprocessing is done in Python and using the public c3d toolbox. Scripts are available on our GitHub.

- **Any other comments?**
  **A:** No.

## 7.5 Uses

- **Has the dataset been used for any tasks already?** If so, please provide a description.
  **A:** We used the dataset for selected benchmark methods we described in our paper. Apart from that, the datasets, that Lung250M-4B builds upon, have been used in several image registration tasks before. We anticipate widespread use by both machine learning researchers in the field of 3D point cloud processing and medical image registration (cf. [12]).

- **Is there a repository that links to any or all papers or systems that use the dataset?** If so, please provide a link or other access point.
  **A:** Not yet, but we plan to enable authors of upcoming methods that use our dataset as a benchmark to link their system on GitHub or paperswithcode.

- **What (other) tasks could the dataset be used for?**
  **A:** The dataset can be used to pre-train either 3D image-based or point-cloud deep learning models in particular for other tasks related to motion and/or highly deformable 3D objects. The derived algorithms can become important tools for medical diagnostics, treatment planning, interactive image-guidance systems and many other things. Research papers on 3D method are number one category of CVPR 2023 `https://public.tableau.com` and we envision a secondary use of our dataset for at least a subset of the methods presented in these papers.

- **Is there anything about the composition of the dataset or the way it was collected and preprocessed/cleaned/labeled that might impact future uses?** For example, is there anything that a dataset consumer might need to know to avoid uses that could result in unfair treatment of individuals or groups (e.g., stereotyping, quality of service issues) or other risks or harms (e.g., legal risks, financial harms)? If so, please provide a description. Is there anything a dataset consumer could do to mitigate these risks or harms?
  **A:** Patients and occurring pathologies are not representative of the general population.

- **Are there tasks for which the dataset should not be used? If so, please provide a description.**
  **A:** This dataset is to be used for research purposes only. It is not intended for clinical usage.

- **Any other comments?**
  **A:** No.

## 7.6 Distribution

- **Will the dataset be distributed to third parties outside of the entity (e.g., company, institution, organization) on behalf of which the dataset was created?** If so, please provide a description.
  **A:** The dataset will be released to the general public but not to any specific third party.

- **How will the dataset be distributed (e.g., tarball on website, API, GitHub)?** Does the dataset have a digital object identifier (DOI)?
  **A:** The image data is currently available through the cloud of the University of Lübeck. The point cloud data is available via Zenodo with a DOI. All code is available on GitHub.

- **When will the dataset be distributed?**
  **A:** The dataset is available immediately.

- **Will the dataset be distributed under a copyright or other intellectual property (IP) license, and/or under applicable terms of use (ToU)?** If so, please describe this license and/or ToU, and provide a link or other access point to, or otherwise reproduce, any relevant licensing terms or ToU, as well as any fees associated with these restrictions.
  **A:** The dataset is distributed under CC-BY-4.0 licence. This excludes CT scans from the DIR-LAB COPDgene dataset and EMPIRE10.

- **Have any third parties imposed IP-based or other restrictions on the data associated with the instances?** If so, please describe these restrictions, and provide a link or other access point to, or otherwise reproduce, any relevant licensing terms, as well as any fees associated with these restrictions.
  **A:** No IP-based restrictions apart from abiding to e.g. referencing original data creators according to CC-BY-4.0 licence guidelines are imposed.

- **Do any export controls or other regulatory restrictions apply to the dataset or to individual instances?** If so, please describe these restrictions, and provide a link or other access point to, or otherwise reproduce, any supporting documentation.
  **A:** No.

- **Any other comments?**
  **A:** No.

## 7.7 Maintenance

- **Who will be supporting/hosting/maintaining the dataset?**
  **A:** Our research group at the University of Lübeck will continue to host and maintain the dataset.

- **How can the owner/curator/manager of the dataset be contacted (e.g., email address)?**
  **A:** Mattias Heinrich can be contacted to communicate queries regarding the dataset `heinrich (at) imi (dot) uni (dash) luebeck (dot) de`

- **Is there an erratum?** If so, please provide a link or other access point.
  **A:** We plan to document possible corrections to the dataset via GitHub.

- **Will the dataset be updated (e.g., to correct labeling errors, add new instances, delete instances)?** If so, please describe how often, by whom, and how updates will be communicated to dataset consumers (e.g., mailing list, GitHub)?
  **A:** We do not plan to regularly update the dataset. However, if it should be necessary, we will communicate this via GitHub.

- **If the dataset relates to people, are there applicable limits on the retention of the data associated with the instances (e.g., were the individuals in question told that their data would be retained for a fixed period of time and then deleted)?** If so, please describe these limits and explain how they will be enforced.
  **A:** No.

- **Will older versions of the dataset continue to be supported/hosted/maintained?** If so, please describe how. If not, please describe how its obsolescence will be communicated to dataset consumers.
  **A:** We do not plan to change the general structure of the dataset even with a possible update, so there should be no need for user customization in this case. In case we do change the

general structure of the dataset, we will provide tools to migrate from the outdated version to the current version. We will document all updates and changes on GitHub.

- **If others want to extend/augment/build on/contribute to the dataset, is there a mechanism for them to do so?** If so, please provide a description. Will these contributions be validated/verified? If so, please describe how. If not, why not? Is there a process for communicating/distributing these contributions to dataset consumers? If so, please provide a description.

  **A:** Interested third parties are welcome to contact us directly to discuss extensions of the dataset. In principle, adding new cases from TCIA to the dataset is as simple as preparing a .csv meta-data file that contains the respective DICOM series IDs. We also plan to implement a mechanism to comprehensibly validate new technical contributions that are applied to the dataset and would implement a leaderboard. If an extension/augmentation/contribution occurs, we will document it via GitHub.

- **Any other comments?**

  **A:** We include a comparison of benchmark results in our GitHub repository. Researchers are welcome to submit their own results on Lung250M-4B. We will verify the correctness of these results before making them publicly available.

## 8 Author statement

As authors, we confirm that we bear all responsibility in case of any violation of rights during the collection of the data or other work, and will take appropriate action when needed, e.g. to remove data with such issues.