# OpenReview forum: "Lung250M-4B: A Combined 3D Dataset for CT- and Point Cloud-Based Intra-Patient Lung Registration"
_NeurIPS.cc/2023/Track/Datasets_and_Benchmarks — NeurIPS 2023 Datasets and Benchmarks Poster_

### Official Review · Reviewer_fFmp · 2023-07-07
**A combined datasets for CT and point cloud registration**

**Rating:** 7
**Confidence:** 3
**Clarity:** Yes, this paper is well organized.

**Strengths:**

The motivation behind creating this dataset, along with the limitations of previous datasets, has been thoroughly addressed. This dataset stands out as the pioneering one to offer a combined collection of image and point cloud data.

Point cloud data is extracted in a reasonable way.

While it's worth noting that the correspondence labels were not generated manually, it is impressive that the trained model still achieved remarkable performance on the test data that was manually labeled.

Additionally, the creators have made valuable contributions by providing preprocessing code, facilitating easy reproduction of their work.

**Additional Feedback:**

(Line 277) 2.24 -> 2.26 ?

Full name of TRE is missing.

**Correctness:**

Need to analyze whether the corresponding labels of the training data are correct. The other parts look fine.

**Documentation:**

Sufficient detail on the dataset is provided.

I can access the URL to the dataset and pre-trained model parameters of the comparison method are also provided.

**Limitations:**

The limitations have been thoroughly addressed and discussed.

**Opportunities For Improvement:**

Could there be a potential hybrid model that effectively leverages both image and point cloud data? Such a model could offer a significant and valuable contribution to the field.

In Table 2, it would be beneficial to include additional comparison methods, particularly focusing on point-based registration methods to provide a more comprehensive evaluation.

It is important to provide a detailed analysis of how corrField serves as a correct label, emphasizing its accuracy and reliability as a reference. Regarding Figure 3 (c), a more elaborate explanation would be beneficial to help readers fully understand the significance and implications of the presented results.

Furthermore, it would be valuable to conduct a comparison with models trained using previous registration datasets to better understand the advancements and improvements achieved by the current dataset and approach.

**Relation To Prior Work:**

Yes, they compared Lung250M-4B with existing datasets and explained the difference.

**Summary And Contributions:**

This paper introduces Lung250M-4B, an innovative dataset comprising both images and point-clouds designed specifically for the intra-patient registration task. To construct this dataset, the authors gathered images exhibiting various ranges of motions from diverse previous datasets. Subsequently, they employed a sophisticated image processing pipeline to extract point clouds from these images. The performance evaluation of Lung250M-4B involved the thorough assessment of existing optimization- and learning-based methods.

---

> ### Author Response · Authors · 2023-08-22
>
> We are very thankful for the reviewer’s comments. We address the raised questions below.
> ### Could there be a potential hybrid model for image/point cloud data?
> This is a very interesting question, since for the development of point-based methods the question of how much image information is still needed or rather how much image information might be redundant, remains of great importance.
>
> In our opinion, combining image and point cloud data in a model could benefit from the accuracy of image-based methods as well as the robustness of point cloud-based models.
> For example, in [1], a point cloud-based registration approach was evaluated using purely geometrical features with or without additional sparsely sampled image-based features. While the method already shows great performance without image features, including image features still improves the registration accuracy.
>
> Since we provide both types of data, our dataset should facilitate the development of such hybrid models. We added lines 109 to 110 to point out this potential in the paper.
> ### How accurate and reliable are corrField labels?
> We chose to compute automatic corrField labels, since they provide very reliable, accurate and relatively dense correspondences for all training pairs. The accuracy was reported in Tab. 2 of the main paper as 1.45 mm (75% 1.77 mm) for the test cases and Tab. 2 of the supplementary as 2.67 mm (75%: 2.42 mm) for the validation set. While less accurate than inter-rater agreement between two manual raters (0.68±1.10 mm), corrField correspondences are automatic and thus come at few extra costs and are significantly denser than sparse manual landmarks. We consider this a good compromise, as it enables us to provide a large set of 700’861 corrField labels in our dataset, compared to 1’700 manual landmarks.
>
> We thank the reviewer for the appreciating observation that models trained on these automatically generated correspondences also achieve very accurate results on the test data with respect to manual landmarks. We hence conclude that the trained models are able to generalise even if partial inaccuracies of keypoints are present and may outperform the provided correspondences in some instances.
> ### How do models trained on Lung250M-4B compare with models trained on previous datasets?
> We performed more experiments to properly address this question and compared alternative training setups with Voxelmorph++ using only singular subsets of our training data.
>
> The results are as follows:
>
> | Training setup | TRE in mm on test data | TRE in mm on validation data |
> | :---- |--|--|
> | TCIA-NSCLC | 5.07 | 4.07 |
> | TCIA-NLST | 3.30 | 3.04 |
> | Empire10 | 3.35 | 3.69 |
> | TCIA-Ventilation | 3.19 | 3.27 |
> | L2R-LungCT | 2.90 | 3.32 |
>
> We added an additional paragraph addressing this (lines 278 to 288) and amended table 3 in the main paper and table 4 in the supplementary material for detailed results including percentiles on the test respectively validation set.
>
> When comparing models trained on our dataset with state-of-the-art results of image-based methods on DIR-LAB COPDgene (like we referenced in line 291), it is important to note that those are usually trained on in-domain DIR-LAB COPDgene cases and partially use manual landmarks for training, whereas we trained and evaluated the benchmarks with a fixed train/validation/test split using DIR-LAB COPDgene solely as test data. Our proposed train/test split however compares well with training on in-domain cases: Training Voxelmorph++ in a 2-fold manner on DIR-LAB COPDgene in addition to the full remaining dataset as training data achieves a TRE of 2.10 mm. We added lines 283 to 287 in the main paper, where we state this result and make the split of training and test data more clear.
>
> For the PVT1010 dataset a fixed train/test split is suggested, with DIR-LAB COPDgene posing as the test data. In the “Comparison to Published Work” paragraph of section 5 of our paper, we refer to published results of comparable methods on the PVT1010 dataset. In the meantime, a preprint [2] of an upcoming paper of authors from our group was published, where one of the comparison methods include a comparable setup of PPWC-Net training on PVT1010, which achieves a TRE of 4.50 mm. We added lines 297 to 299 in our paper to refer to these results.
> ### Additional comparison methods
> We agree that a good overview over results of different methods is relevant and included a results table as part of our GitHub repository that we plan to keep updated with results of different models employed on the dataset.
>
> Again, we appreciate the detailed feedback. If there are any more questions, please let us know.
>
> [1] Hansen, L. & Heinrich, M. P. (2021). Deep learning based geometric registration for medical images: How accurate can we get without visual features? IPMI 2021.
>
> [2] Bigalke, A. & Heinrich, M. P. (2023). A denoised Mean Teacher for domain adaptive point cloud registration. arXiv preprint arXiv:2306.14749.

---

### Official Review · Reviewer_D1Xr · 2023-07-20
**A new benchmark 3D dataset for lung registration**

**Rating:** 7
**Confidence:** 3
**Clarity:** The paper is well-written and easy to…

**Strengths:**

* The proposed dataset is larger and more comprehensive than existing datasets, which can be a valuable resource for researchers in the field of medical image registration.
* The authors provide a sufficient description of the methods used for dataset generation, including image processing, vessel and point cloud extraction, and automatic correspondences.
* The authors have conducted a comprehensive evaluation of different registration methods on the dataset, which provides a benchmark for future research in this area.

**Additional Feedback:**

No

**Correctness:**

The claims seem to be correct. The authors provides a detailed description of how the dataset was constructed which is reasonable. The also provides code and implementation details for evaluating the results.

**Documentation:**

Yes

**Limitations:**

* The authors acknowledge a few technical limitations that could affect the applicability and generalizability of the dataset.
* The authors are expected to discuss the potential misuse or unintended consequences of their work.
* It is helpful to discuss the potential population biases in the dataset and how these biases would affect the results.

**Opportunities For Improvement:**

* The dataset is relatively small, although it is larger than exisitng ones. It is helpful to discuss how the proposed dataset could be used in conjunction with existing ones to address a wider range of research questions.
* It is helpful to include more comparative methods for a more comprehensive evlaution on the dataset.

**Relation To Prior Work:**

The authors discussed how their dataset differs from existing ones regarding the methods used to extract point clouds and generate labels and correspondences. However, it is better to provide a more detailed comparison with previous datasets and methods. For examples, how the specific challenges of intra-patient lung registration differ from those addressed by existing datasets and methods. They could also provide a more detailed comparison of the methods used to construct their dataset with those used in previous work.

**Summary And Contributions:**

This study presents Lung250M-4B, a novel 3D dataset specifically created for intra-patient lung registration. The dataset aims to overcome the shortcomings of current datasets, such as inadequate sample size and limited range of motion. Comprising 248 paired CT scans from 124 patients, along with corresponding lung vessel point clouds, Lung250M-4B provides a comprehensive resource for this field of study. The authors carry out an evaluation of various registration methods using this dataset. The authors have made the Lung250M-4B dataset publicly accessible, and have also shared all relevant source code, tools, scripts, and models necessary for the preprocessing stages.

---

> ### Author Response · Authors · 2023-08-22
>
> We are very thankful for the reviewer’s comments. We address the raised questions below.
> ### How could the proposed dataset be used in conjunction with existing ones?
> Our dataset can be easily and individually expanded with more data, including private datasets, by the provided preprocessing scripts. How the dataset is generated is made completely open-source and is easily adaptable (by changing path and file names) to other datasets. We consider this to be a main strength of the dataset.
>
> ### Bias and misuse
> We address potential biases in our dataset as well as potential misuse in the supplementary material (cf. Tab. 1 and section 7.5). To make this also clear in the main paper, we added lines 183 to 185 in the ethical discussion part of the main paper.
>
> The main bias present in the dataset lies in the occurrence of pathologies, which does not reflect the general population. This results from (i) CT scans usually being acquired following a medical indication and (ii) the data comprising multiple sub-datasets acquired during different studies with a focus on different diagnoses (cf. Suppl. Tab. 1 for more details). Thus, as stated in the supplementary material and now also in the main paper, the dataset is not directly intended for training algorithms for clinical use. However, for methodological development, we expect the variety of occurring pathologies in our dataset to be beneficial rather than disadvantageous.
>
> Because we curate data from different sub-datasets, our dataset includes data from patients with varying pathologies and we can reduce ethnic and demographic biases that may be present in singular sub-datasets. We consider it important to be aware of such biases in the dataset, which is why we included the detailed breakdown of all sub-datasets and information that is available in this regard in Tab. 1 of the Supplementary Material.
>
> ### How does intra-patient lung registration differ from similar tasks?
> Intra-patient lung registration stands out for large deformation of small anatomical structures, which poses a difficult problem for learning-based methods. Furthermore, with a lot of branched tree structures inside of the lung (i.e. lung vessels), that are well suited for graphs, it offers great potential for point-based methods. As seen in the Learn2Reg [1] challenges over the years, in comparison to other tasks, the intra-patient lung registration task continually poses as very competitive between learning-based and conventional methods as well as point cloud- and voxel-based methods. We consider the Lung250M-4B dataset to be of great value to fairly compare these methods against each other and further the development of learning-based methods that heavily rely on sufficient training data.
>
> In general, we consider intra-patient lung registration to be a task of high clinical relevance with use cases including radiotherapy and lung nodule tracking (cf. section 1, paragraph “Clinical Motivation for Lung Registration”).
> ### Comparison between methods used to generate our dataset and other datasets
> A possible comparison can be made with the PVT1010 dataset [2]. There, they first used a Frangi filter to obtain vessel-like structure in the data and used a refinement on that result in order to obtain relevant points that are located around the vessel centerlines. Instead of the Frangi filter, we used the vesselness network, which, according to the authors' publication [3], outperforms the Frangi filter in most cases. Additionally, different from PVT1010, we use these segmentations alongside ground truth segmentations (for the arteries) to train a network, which bears a higher generalisation potential. Lastly, through the skeletonisation, we also obtain points sampled along the centerline.
> ### Inclusion of more comparative methods
> We agree that a good overview regarding results of different methods is relevant and included a results table as part of our GitHub repository that we plan to keep updated with results of different models employed on the dataset. We also added some additional experiments to our paper (as stated in our general comment).
>
> Again, we appreciate the detailed feedback. If there are any more questions, please let us know.
>
> [1] Hering, A. et al. (2022). Learn2Reg: comprehensive multi-task medical image registration challenge, dataset and evaluation in the era of deep learning. IEEE Transactions on Medical Imaging, 42(3), 697-712.
>
> [2] Shen, Z. et al. (2021). Accurate point cloud registration with robust optimal transport. Advances in Neural Information Processing Systems, 34, 5373-5389.
>
> [3] Jena, R., Singla, S., & Batmanghelich, K. (2021). Self-supervised vessel enhancement using flow-based consistencies. In Medical Image Computing and Computer Assisted Intervention–MICCAI 2021: 24th International Conference, Strasbourg, France, September 27–October 1, 2021, Proceedings, Part II 24 (pp. 242-251). Springer International Publishing.

---

> > ### Comment · Reviewer_D1Xr · 2023-08-29
> >
> > I would like to thank authors for addressing my comments. This is a great work. I changed my score from 6 to 7.

---

### Official Review · Reviewer_gUyw · 2023-07-21
**Lung250M-4B: A Combined 3D Dataset for CT- and Point Cloud-Based Intra-Patient Lung Registration**

**Rating:** 9
**Confidence:** 4
**Correctness:** Good
**Clarity:** Yes

**Strengths:**

Very clearly written.

Reasonably chosen algorithms to benchmark.

Clearly described methods on how the dataset was produced.

**Additional Feedback:**

None

**Documentation:**

Clear github repo and documentation.


**Ethics:**

Good ethical review.

**Limitations:**

None.



**Opportunities For Improvement:**

No criticisms as such.

The description of point correspondence could be expanded: The title suggests that the dataset is primarily for intra-patient (inhale/exhale) registration. But ML methods would require the same number of points in inhale/exhale. I can see how their methodology would work for each patient, but not inter-patient? Can you guarantee corresponding points between patients? I would imagine the anatomy is too variable. So, can you either either explain how you can do it, or how it is not possible and why.

Page 2, line 34, 35: Quoting the number of voxels and number of 3D points is to my mind, a bit unnecessary, unhelpful and even misleading. It's more clear, to just say data from 124 patients, inhale/exhale so 248 scans as that is more relevant.







**Relation To Prior Work:**

Well explained.


**Summary And Contributions:**

The authors provide a dataset and some benchmark performance figures for research into Intra-Patient Lung Registration. The major contribution is that this is the first dataset to provide both CT data and corresponding point-cloud data from the lung vessels.

The dataset is 3-4 times larger than existing datasets, and the only dataset to provide combined image+points from the same patients.

Thus this is the first to also provide benchmarks of common algorithms, taking into account image and point base methods, both learning and classical methods, and unsupervised/self-supervised and supervised.

---

> ### Author Response · Authors · 2023-08-22
>
> We are very thankful for the reviewer’s comments and would like to address the question regarding the point correspondences between patients and consequent applicability of this dataset for inter-patient lung registration.
>
> Going along with the current trends and ongoing research in the community, we focus on intra-patient registration with this dataset. Due to this, we decided to include annotations that correspond within each in-/exhale scan pair, but not necessarily across patients. While this unfortunately reduces the usefulness of the annotations for inter-patient registration, this has some advantages, i.a.:
> - We were able to include the official DIR-LAB COPDgene landmarks, which do not correspond between patients, for testing.
> - We could generate point clouds based on vessel structures, which provide a lot of information for intra-patient registration, but are not very informative for inter-patient registration, since vessel topology differs greatly across different patients.
>
> Furthermore, since inter-patient registration does not rely on intra-patient pairs of in- and expiratory phases, a lot more existing lung CT datasets would come into question for this application.
> However, an inclusion of inter-patient corresponding annotations, e.g. of airway structures, could be interesting.
>
> Again, we appreciate the detailed feedback. If there are any more questions, please let us know.

---

> > ### Comment · Reviewer_gUyw · 2023-08-29
> > **Thanks**
> >
> > Thanks for your response. Reading these comments, and other comments to other reviewers, I believe you've done a good job responding. I'll leave my score as-is, as I think it's a fair score, and was positive in the first place.

---

### Official Review · Reviewer_yrwe · 2023-07-24
**Review of Lung250M-4B**

**Rating:** 5
**Confidence:** 4
**Correctness:** Yes
**Clarity:** Yes

**Strengths:**

- Excellent Writing: The submission is well-written and easy to comprehend. The authors clearly explain the limitations of existing datasets and detail their methodology in an organized manner. This makes it easy for readers to understand the value of their contribution and the steps they took to curate this new dataset.

- Large-scale Dataset: The authors curated a comprehensive and large-scale dataset, which is a significant contribution in itself. They have processed 248 lung CT scans from 124 patients, making it one of the largest public datasets for lung registration. The inclusion of both small and large motion scans increases the applicability of the dataset across a range of scenarios.

- Holistic Approach: The authors go beyond data curation and offer both training and validation tools, including vein and artery segmentations, multiple thousand image-derived keypoint correspondences, and manual landmark annotations. This holistic approach will be valuable for researchers and practitioners looking to train or evaluate their lung registration models.

- Benchmarking: The authors also take the initiative to evaluate several image and point cloud registration methods on their dataset, providing a valuable starting point for future researchers. This helps establish their dataset as a robust benchmark for lung registration tasks.

**Additional Feedback:**

NA

**Documentation:**

Yes

**Ethics:**

Yes

**Limitations:**

Yes

**Opportunities For Improvement:**

- The authors are using many algorithms to provide auxiliary annotations, such as mask, how to avoid cumulative errors.
- More detailed information could be clarified in the paper. How do you ensure consistency of labeling between different datasets? Is the annotation using isimatch completely correct? Is there any validation?

**Relation To Prior Work:**

Yes

**Summary And Contributions:**

The current benchmark for intra-patient lung registration, the DIR-LAB COPDgene dataset, lacks sufficient sample sizes and includes primarily small motions. Similarly, the PVT1010 dataset for point-based geometric registration lacks correspondences for supervision and public CT scans, making fair comparison with image-based algorithms infeasible. To address these limitations, the authors curate a combined benchmark for both image- and point-based registration approaches, comprising of 248 public multi-centric in- and expiratory lung CT scans from 124 patients. These scans exhibit large motion, and have been processed for homogeneity and generation of well-distributed vessel point clouds. Vein and artery segmentations, along with multiple thousand image-derived keypoint correspondences for each pair, are provided for supervised training. For validation, multiple scan pairs with manual landmark annotations are included. Lastly, the authors evaluate various image and point cloud registration methods on this new dataset as initial baselines.

Key Contributions:

- Data Curation: Collection of 248 public multi-centric in- and expiratory lung CT scans from 124 patients exhibiting large motion.
- Pre-processing: Scans have been processed for homogeneity and for generation of well-distributed vessel point clouds.
- Training & Validation: Provision of vein and artery segmentations, image-derived keypoint correspondences for supervised training, and manual landmark annotations for validation.
- Benchmarking: Initial evaluation of various image and point cloud registration methods to set baselines on the new dataset.

---

> ### Author Response · Authors · 2023-08-22
>
> We are very thankful for the reviewer’s comments. We address the raised questions below.
> ### How do we avoid cumulative errors in annotations?
> One of the annotations we use consecutively are automatically generated binary lung masks. We are aware that the correctness of these is crucial for downstream annotations. We are confident about their quality, as we employed an nnUNet, which is very reliable and accurate for lung mask segmentation, and verified the generated segmentations by visual inspection. The nnUNet framework is generally considered state-of-the-art for semantic segmentation of medical images, winning multiple segmentation challenges for a wide variety of tasks. Additionally, lung mask segmentation is considered a relatively easy task. Since the areas to be segmented are sufficiently large and there are no holes inside of the lung lobes, the likeliest area to be inaccurate is at the boundary of the lung, which has a distinctive edge in CT scans and is thus easy to detect. Furthermore, the vessel segmentations are located far enough inside the lung such that a small inaccuracy would not result in a consecutive error.
>
> Generating the vessel segmentations also consists of multiple steps, as we first generate training labels and then employ a nnUNet segmentation. While the vessel segmentations themselves are automatically generated (similar to the PVT1010 dataset, where a Frangi filter is employed), marking of the arterial parts of the vessels is done based on ground truth, manually refined segmentations from the Parse22 dataset. The validation Dice score for the resulting nnUNet model on leave-out validation cases from the Parse22 dataset (with semantic segmentations generated by us as described) is 81.4% for the vein and 84.9% for the artery. We added this to our paper in line 211 to 213.
>
> ### How do we ensure consistency of labelling between different datasets?
> Since registration is not done inter-patient, keypoint and landmark annotations are generated in a way that they match for each intra-patient scan pair and due to the nature of the annotations generally not 1-to-1 between different patients. Nonetheless, we consider it important to omit systemic errors or biases in the labelling depending on the sub-datasets, in particular for vessel annotations.
>
>
> Landmarks:
> The semi-automatic annotation method, i.e., finding a correspondence in image B to a destined point in image A, is robust to systematic label errors that can occur in other annotation tasks such as classification or segmentation. We would also like to emphasise that since our dataset is aimed to be used for intra-patient registration, labelling inconsistency between different patients within a reasonable range has little impact. However, to mitigate risks of label inconsistencies, we limited ourselves to a small group of annotators and performed a standardised, detailed briefing before labelling.
>
> Vessel annotations:
> Vessel segmentations are generated using a trained nnUNet segmentation network. To support the generalisation ability of the network, we aimed to reduce the domain gap between datasets as far as possible without distorting image information by employing a joint pre-processing.
> Additionally, the topology of vessel tree structures differ between patients, which consequently results in differences between the segmentation topologies in themselves. For intra-patient registration, the consistency between the labels of each image pair remains most important. Those pairs, however, come from the same dataset.
>
> ### Is the annotation using isimatch completely correct? Is there any validation?
> To assess the quality of the annotations, we performed an additional experiment on inter-rater agreement within our annotation setup (same annotators and software) on 400 landmarks in 4 cases and obtain a mean inter-observer error of 0.68 mm, which is less than the image resolution and on par with the inter-observer error of the annotations of the DIR-LAB COPDgene dataset [1]. We report this in the supplementary material (lines 160 to 163).
> We further would like to state that absolute correctness of landmark matching is, at least with discretised image data, not always feasible, since solutions in neighbouring voxels might be equally correct.
>
> Again, we appreciate the detailed feedback. If there are any more questions, please let us know.
>
> [1] Castillo, R., Castillo, E., Fuentes, D., Ahmad, M., Wood, A. M., Ludwig, M. S., & Guerrero, T. (2013). A reference dataset for deformable image registration spatial accuracy evaluation using the COPDgene study archive. Physics in Medicine & Biology, 58(9), 2861.

---

### Author Response · Authors · 2023-08-22
**General comment**

We would like to thank the reviewers for their valuable comments and feedback. We greatly appreciate that the reviewers acknowledged the value of our contribution, especially in terms of focussed application, comprehensive benchmarking, size of the dataset, reproducibility and the inclusion of joint image and point cloud data.

Based on the feedback we received, the following changes were made to the paper:
- To verify the positive influence of our increased dataset size, we performed additional experiments using Voxelmorph++ with different training setups, namely training on only the sub-datasets and training on DIR-Lab COPDgene in a 2-fold cross-validation manner. We added a paragraph that describes these experiments (lines 278 to 282 in the main paper, lines 282 to 286 in the supplementary material). The results are displayed in Tab. 3 in the main paper and Tab. 4 in the supplementary material.
- To demonstrate the quality of automatic vessel segmentation, we computed the validation Dice of the trained nnUNet model (81.4% for the vein, 84.9% for the artery), as reported in lines 211 to 213.
- We added an additional recently published result in the paragraph “Comparison to Published Work” in section 5 (lines 297-299).
- We added an estimation of inter-observer error in the supplement (lines 160 to 163) and specified the number of manually annotated landmark correspondences per case for each subset in Tab. 1 of the main paper.
- We added a remark regarding clinical usage and potential bias in the dataset to the ethical discussion of the main paper (lines 183 to 185).
- We now mention that the inclusion of joint image and point cloud data in our dataset is beneficial for the development of hybrid models that make use of both image and point cloud-data (lines 109 to 110).
- Since we consider a good overview over results of different methods relevant, we included a results table as part of our GitHub repository, which we will keep updated with results of different models employed on the dataset. We added a remark in this regard to section 7.7 of the supplementary material (lines 589 to 591) and prominent in our GitHub repository.
- Lastly, we fixed some minor typographic errors.

Text added in the paper is marked in red.

---

### Decision · Program_Chairs · 2023-09-22

**Decision:**

Accept (Poster)

**Comment:**

This paper presents an innovative dataset addressing the limitations of existing datasets by providing both images and point cloud data for intra-patient lung registration. While there are concerns regarding dataset size, potential hybrid models, and the accuracy of correspondence labels, the paper's strengths lie in its clear motivation for the creation of such a dataset, the approach to point cloud extraction, comprehensive performance evaluation, and the provision of reproducible code. I believe that the positive aspects outweigh the weaknesses, making it a valuable contribution to medical image analysis community, justifying an "accept" decision, especially given the authors' effective responses to reviewer feedback and improvements made in response to their feedback.